# Integrative analysis of transcriptome dynamics during human craniofacial development identifies candidate disease genes

Tara N. Yankee[1], Sungryong Oh [2], Emma Wentworth Winchester [1], Andrea Wilderman[1], Kelsey Robinson[3], Tia Gordon[4], Jill A. Rosenfeld [4,5], Jennifer VanOudenhove [2], Daryl A. Scott [4,6], Elizabeth J. Leslie [3] & Justin Cotney [2,7] ✉

Craniofacial disorders arise in early pregnancy and are one of the most common congenital defects. To fully understand how craniofacial disorders arise, it is essential to characterize gene expression during the patterning of the craniofacial region. To address this, we performed bulk and single-cell RNA-seq on human craniofacial tissue from 4-8 weeks post conception. Comparisons to dozens of other human tissues revealed 239 genes most strongly expressed during craniofacial development. Craniofacial-biased developmental enhancers were enriched +/− 400 kb surrounding these craniofacial-biased genes. Gene co-expression analysis revealed that regulatory hubs are enriched for known disease causing genes and are resistant to mutation in the normal healthy population. Combining transcriptomic and epigenomic data we identified 539 genes likely to contribute to craniofacial disorders. While most have not been previously implicated in craniofacial disorders, we demonstrate this set of genes has increased levels of de novo mutations in orofacial clefting patients warranting further study.

Craniofacial disorders are among the most common of all congenital defects, with clefts of the lip and/or palate being the most frequent, affecting nearly 1 in 700 live births worldwide[1]. These defects are a significant public health issue with far-reaching economic ramifications. For children born with orofacial clefts alone, the combined lifetime cost of treatment is nearly $700 million[2–4]. Craniofacial disorders also pose unique physiological and psychological challenges to patients and their families. The face contains most of the major sensory organs and is the primary means by which humans communicate emotions, a fundamental social behavior necessary to form and cultivate relationships. For these reasons, there is significant impetus to understand the etiology of craniofacial defects to aid in the development of improved diagnostic and preventative methods.

The link between craniofacial abnormalities and genetic disorders has been long established, as over 500 Mendelian syndromes, according to OMIM, exhibit a craniofacial element[5]. Machine learning technology has been able to take advantage of this observation, and several tools have been developed to match facial phenotypes with

[1]Graduate Program in Genetics and Developmental Biology, UConn Health, Farmington, CT 06030, USA. [2]University of Connecticut School of Medicine, Department of Genetics and Genome Sciences, Farmington, CT 06030, USA. [3]Department of Human Genetics, Emory University School of Medicine, Atlanta, GA 30322, USA. [4]Department of Molecular and Human Genetics, Baylor College of Medicine, Houston, TX 77030, USA. [5]Baylor Genetics Laboratory, Houston, TX 77021, USA. [6]Department of Molecular Physiology and Biophysics, Baylor College of Medicine, Houston, TX 77030, USA. [7]Institute for Systems Genomics, University of Connecticut, Storrs, CT 06269, USA. ✉e-mail: cotney@uchc.edu

known syndromes[6-9]. Most recently, machine vision tools have shown potential for applications in diagnosing and characterizing ultra-rare syndromes with the added benefit of being minimally invasive[10,11]. However, the majority of craniofacial abnormalities are nonsyndromic, occurring in the absence of other abnormalities, potentially limiting the utility of such approaches for diagnosing nonsyndromic cases. Additionally, the underlying genetic causes of many nonsyndromic craniofacial abnormalities remain unclear and are likely to be more varied across individuals than the rare syndromes. Genome-wide association studies (GWAS) have identified dozens of loci that contribute to the heritability of nonsyndromic orofacial clefting[12-19] as well as metrics of normal human facial variation[20-22].

Only a handful of these variants reside in genes. The vast majority of these GWAS variants are found in noncoding portions of the genome. We and others have shown that such variants are systematically enriched in sequences that obtain active chromatin states during early human craniofacial development, particularly those annotated as strong enhancers[23-25]. Strong enhancers only active in developing craniofacial tissues are enriched for craniofacially relevant transcription factor (TF) binding sites and are generally located nearby genes already implicated in craniofacial abnormalities. However, these enrichments are driven by a relatively small fraction of all identified craniofacial enhancers, and it is unclear exactly which genes these enhancers regulate. Establishing such enhancer-gene assignments is difficult in the absence of direct gene expression measurements[26-28]. Thus far, gene expression during craniofacial development has been primarily studied in model organisms such as mouse, chicken, and zebrafish[29-32]. For ethical and logistical reasons, gene expression data from primary craniofacial developing human tissue is relatively rare[33,34] compared to the thousands of gene expression datasets produced in adult tissues[35,36]. Thus there is a great need for large-scale genome-wide data in order to comprehensively study human craniofacial development and better understand which genes are regulated by the trait loci identified in patient cohorts.

To address this knowledge gap, we have generated an extensive set of bulk RNA-seq data from organogenesis of the human face. Here we describe the analysis of these gene expression data along with the integration of thousands of gene expression profiles from adult tissues and our previously published craniofacial chromatin state data[25]. These analyses revealed genes with previously unappreciated craniofacial-biased expression. We combined these findings with data on craniofacial-specific enhancers to predict enhancer:gene pairs in the developing human face. We also characterized gene coexpression across developmental time, revealing gene coexpression modules that are strongly enriched for known disease-causing genes. These results provide a framework for prioritizing putative disease genes that share similar features across our data. This prioritization strategy identifies 539 genes that are significantly enriched for de novo protein-altering mutations in patients with orofacial clefting. Using large public repositories of genome sequencing (GS) data from healthy individuals, and sequence and copy number variation data from individuals with developmental abnormalities, we show strong evidence that haploinsufficiency of one of our predicted genes, *EBF3*, is a risk factor for orofacial clefting in humans. Using single nucleus gene expression data from comparable timepoints in human and mouse craniofacial development, we show distinct differences in the expression of *EBF3* and its paralog *EBF2* in multiple populations of mesenchyme cells across species. Data from this investigation are available from the Genome Expression Omnibus, dbGAP, as a public track hub on the University of California Santa Cruz (UCSC) Genome Browser, and directly from our laboratory website for convenient exploration by the field (https://cotney.research.uchc.edu/craniofacial).

## Results

### Characterization of global gene expression during human craniofacial development

Gene expression during human organogenesis has been shown to be highly dynamic, and its precise orchestration is crucial for the normal patterning of a variety of tissues and structures[37-39]. However, gene expression patterns during critical stages of human craniofacial development have not been systematically profiled. To characterize the transcriptome of the developing human face, we generated gene expression profiles from multiple biological replicates of primary craniofacial (CF) tissue from four distinct Carnegie Stages (CS) of the embryonic period (CS13, CS14, CS15, and CS17) as indicated (Fig. 1a). This is equivalent to approximately 4 to 6 post-conception weeks (pcw) based on established staging criteria[40]. We also retrieved CS22 (~8 pcw) gene expression data from Facebase[34,41], resulting in a time series dataset encompassing nearly half of the embryonic period. We have previously characterized chromatin state dynamics during these same stages, including six of these same samples (Supplementary Data 1), and showed that regulatory regions active in the face during this period harbor much of the risk for orofacial clefting identified by GWAS[25]. Quality control metrics of this bulk gene expression data revealed that newly generated samples had minimal 3' bias, generally high RNA integrity number (RIN) values and excellent transcript integrity number (TIN) scores (>70) (Fig. 1b, Supplementary Fig. 1A, and Supplementary Data 1). Principal component analysis (PCA) of the CF samples showed minimal effects from sex and RNA quality in the first 2 components (Supplementary Fig. 1B). Having confirmed the high quality of these samples, we then summarized gene expression at the composite gene level and identified an average of 26,000 genes expressed at each timepoint (see Methods). The greatest number of genes were expressed at the latest timepoints of the developmental series. This is consistent with the increasing heterogeneity of tissues and cell types generated as development progresses. These different tissues are undoubtedly present in these bulk samples.

While these timepoints may capture many of the critical events in CF patterning, other events that occur before CS13 are difficult to address, and targeted experiments cannot be performed in vivo. Recent advances in directed differentiation of hESCs into different cell types have resulted in cell populations that recapitulate many of the properties of cranial neural crest cells (CNCCs), which contribute to the developing face[42-44]. However, the similarity of cultured CNCC transcriptomes to the primary tissue has not been definitively demonstrated. To address these issues, we generated CNCCs from H9 hESCs across three distinct rounds of differentiation using an established protocol[44] and profiled global gene expression. To systematically determine the similarity of these CNCC culture systems, we performed a PCA of these cultured CNCC samples with the primary CF samples. This revealed that the first principal component reflected the expected developmental time (Fig. 1b). To further verify the relevance of the CNCC culture model, we performed differential gene expression analysis between H9 hESCs and CNCCs as well as CS13 samples. We identified 2476 and 3740 genes upregulated versus hESCs in CNCCs and CS13 samples, respectively. We found a significant (permutation $P < 0.001$) overlap of 1710 upregulated genes between these two comparisons. Gene ontology of these genes revealed many terms that were relevant to craniofacial development and indicated that the CNCC differentiation captured many of the expression changes in the bulk tissue compared to pluripotent stem cells (Supplementary Fig. 1C).

While these initial comparisons gave us a sense of the overall trends in the data, they did not yet reveal clear craniofacial-relevant biology. To gain more specific insight into craniofacial gene expression patterns, we reasoned that comparisons with many other tissues could reveal such information. To achieve this, we uniformly processed our primary human embryonic and CNCC data with the *Rail-RNA* pipeline[45]

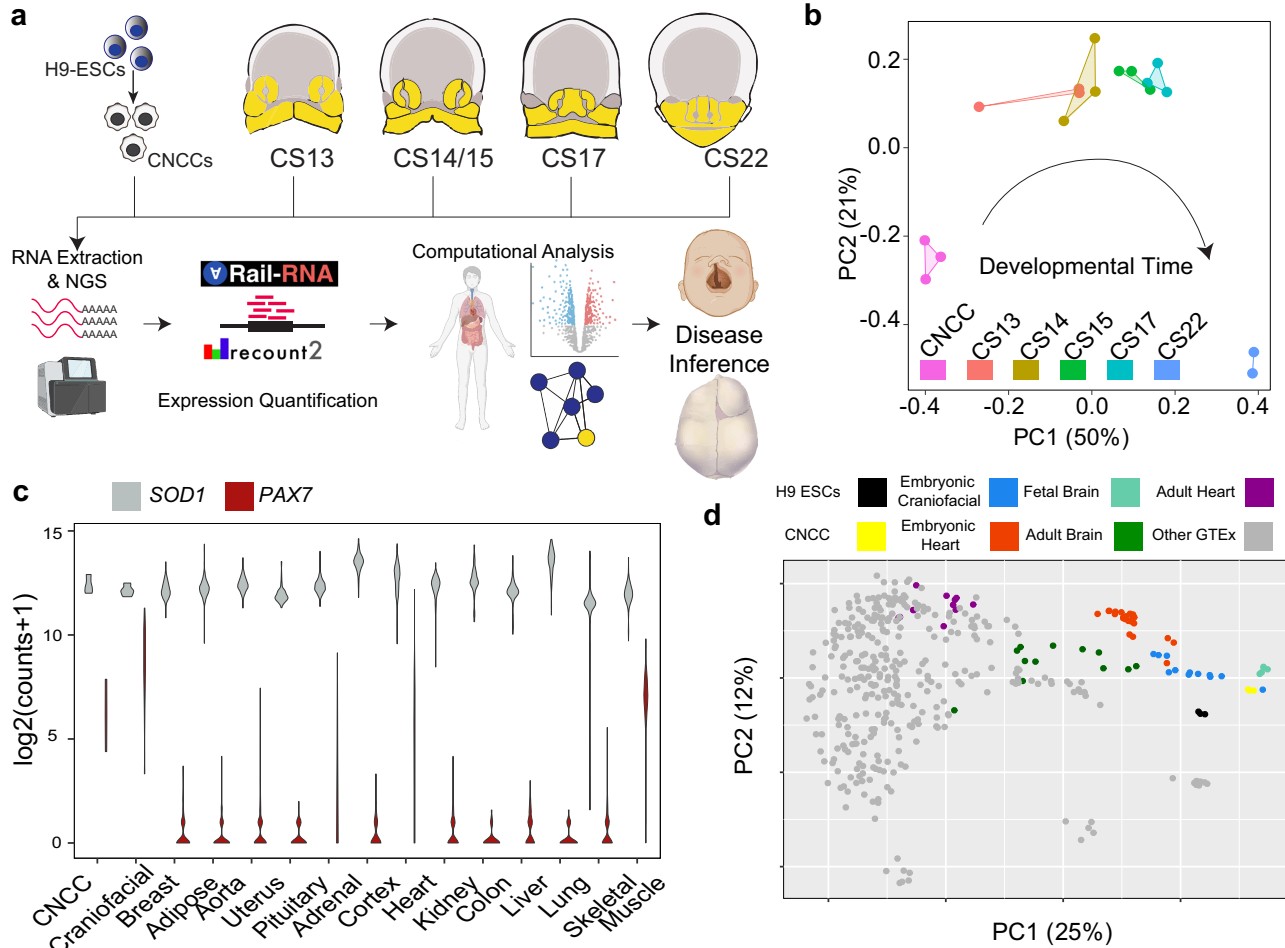

**Fig. 1 | Transcriptomic profiling of early human craniofacial development.**
**a** Overview of experimental design. Bulk mRNA-seq was performed on five-timepoints of the primary CF tissue and a cell culture model of CNCCs. The Rail-RNA/recount2 pipeline was used for alignment and gene-level transcript quantification. In order to determine potential disease genes within craniofacial development, several downstream computational analyses were performed, such as multi-tissue comparisons, pairwise differential expression, and the generation of coexpression networks. Created with BioRender.com. **b** PCA plot showing samples (identified by color) vary the most by time which is captured by PC1. **c** Violin plots of *PAX7* and *SOD1* expression across all samples per tissue surveyed. Comparison of *PAX7* between CNCC and craniofacial using a two-sided Mann-Whitney test showed a non-significant difference. **d** PCA plot of the top 5000 most variable genes between H9 ESCs, H9 derived CNCCs, the early primary craniofacial tissue, embryonic heart, fetal brain, and 12 random samples from each of the primary tissues available in GTEx.

employed by the *recount2* database[46]. This enabled direct comparisons of our gene expression data to thousands of samples collected from tissues profiled by GTEx[36], multiple stages of the developing human brain[47], and samples we previously generated from the developing human heart[39]. We compared composite gene level expression across all of these samples and observed similar numbers of expressed genes and ranges of gene expression (Fig. 1d). When we inspected the developmental transcription factor *PAX7*, known to be expressed in early neural crest and many of the craniofacial derivatives[48] we observed robust expression biased toward CNCC and CF samples relative to other tissues (Fig. 1c and Supplementary Fig. 1E). This is in contrast to *SOD1*, which is frequently used as a housekeeping control gene, and was expressed at similar levels in all tissues (Fig. 1c and Supplementary Fig. 1E).

We then performed a t-distributed stochastic neighbor embedding (tSNE)[49] analysis on all expressed genes across all tissue samples and found a distinct cluster of craniofacial samples, along with clusters for each of the tissues profiled by GTEx (Supplementary Fig. 1F). This trend was also observed in PCA, which showed CF tissues generally clustered together (Fig. 1d). Overall, these results demonstrate the quality of our data, the robustness of individual embryo staging, the

validity of the CNCC system as a model for early craniofacial development, and recovery of a clear developmental time component in our gene expression data. With these global characteristics confirmed, we set out to perform specific analyses directed at identifying genes specifically expressed in these CF tissue samples, genes differentially expressed across this developmental trajectory, and the construction of networks of coexpressed genes, which we detail below.

## Genes expressed specifically during craniofacial development are enriched for craniofacial abnormalities

The specificity of gene expression is an important aspect of cell identity in multicellular organisms. Restricted gene expression is frequently observed for genes directly related to tissue-specific functionality and is often an indicator of disease relevance[50–52]. Our previous work has shown that strongly biased expression directly identified known genes for congenital heart defects[39]. We looked to do a similar analysis with this craniofacial dataset to identify putative CF disease genes; as demonstrated above, *PAX7* expression seemed to be strongly biased toward our CF samples.

To systematically identify genes with similar patterns of tissue-biased expression, we used a measurement of inequality called the Gini

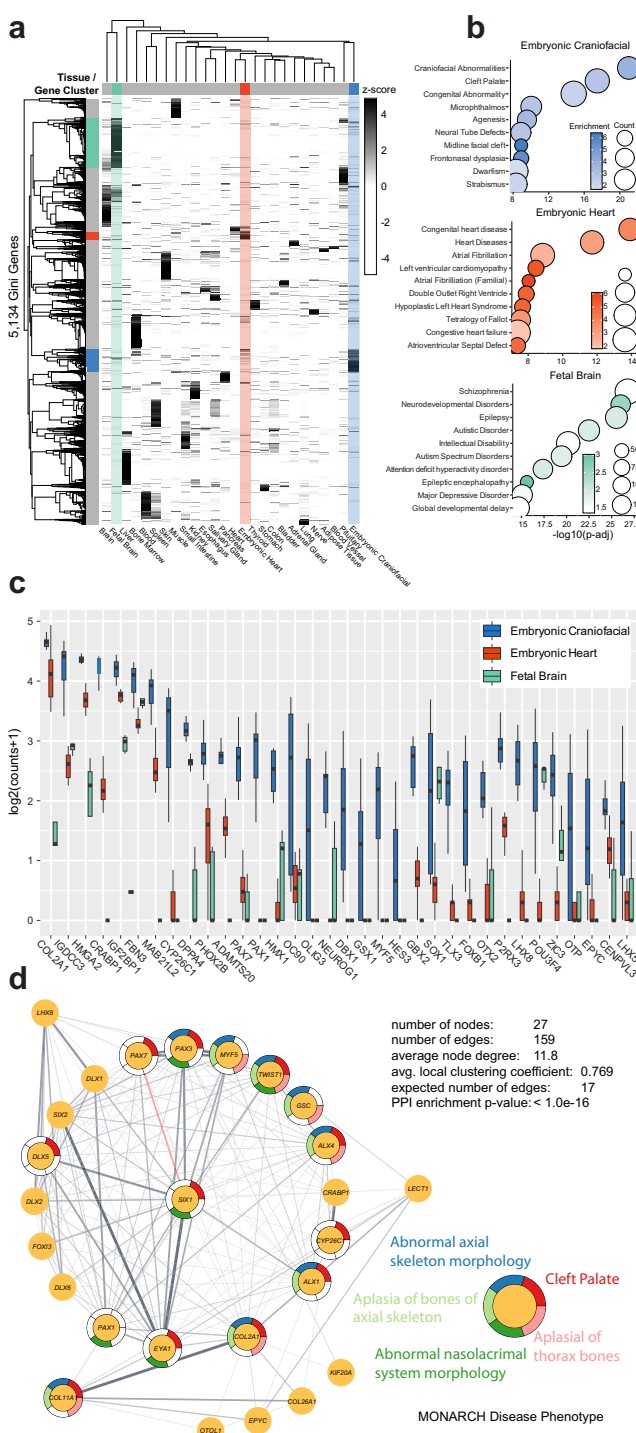

**Fig. 2 | Gini Index unbiasedly identifies gene expression specific to embryonic craniofacial tissue genome-wide. a** Heatmap showing the specificity of expression for tissue-specific genes (rows) identified with high Gini scores (≥0.7) for all non-sex (n = 25) tissues (columns) from GTEx, embryonic heart, and fetal brain. Gene expression increases from light to dark. Embryonic craniofacial (blue), embryonic heart (red), and fetal brain (light green) specific genes and samples are highlighted, all other tissues are marked gray. **b** Selected significant (Benjamini and Hochberg–adjusted p value <0.05) disease ontology enrichments for genes identified to be specific for these tissues (Gini ≥0.7 and highest average expression). The dot size or count indicates the number of genes within a disease category. Color bars are log2 of the fold enrichment (see Methods). **c** Boxplots of gene expression of highest scoring Gini genes for the craniofacial tissues (n = 12) compared to embryonic heart (n = 24) and fetal brain (n = 5), of which all samples are biologically independent. The center line denotes the median value (50th percentile), the box contains the 25th to 75th percentiles and the whiskers mark the 5th and 95th percentiles. Data points beyond these values (outliers) are not shown. **d** Protein–protein interactions of 27 genes with conserved craniofacial expression bias in both humans and mouse as calculated by StringDB[155]. The outer ring for each node indicates membership in MONARCH[156]. Disease ontology categories were identified as significantly enriched in this gene set.

Index across all tissues profiled above[53–55] (Supplementary Data 2). This index results in a bimodal distribution of genes with a median value of 0.42 and a large number of genes with Gini scores ranging from 0 to 0.3. These low-scoring genes are not tissue-specific, and are thus expressed across many cell types and likely housekeeping genes (e.g., *SOD1* = 0.238). Notably, several genes involved in the craniofacial-biased disease Treacher Collins syndrome, *TCOF1*, *POLR1C*, and *POLR1D*, were low-scoring genes found to be expressed at high levels in virtually all tissues (Gini scores 0.23, 0.19, and 0.15 respectively). The top quartile of genes, those with scores of 0.7 or greater, contained 5,134 "Gini" genes that have strongly biased expression towards a small number of samples (Fig. 2a). Of these biased genes, 239 genes demonstrated the highest average expression in our CF samples. To

determine the CF disease relevance of these genes, we performed a disease enrichment analysis of these CF Gini genes using the DisGeNET database[56] and found they were indeed most significantly enriched for genes implicated in CF abnormalities and cleft palate. Similarly, fetal brain and embryonic heart Gini genes were significantly enriched for neurological disease and congenital heart defect-related ontologies, respectively (Fig. 2b). In contrast to the Gini index analysis, using the same number of genes ranked solely by absolute expression in craniofacial samples we observed gene ontology enrichments for general cellular and metabolic processes and distinct disease enrichments including Diamond Blackfan anemia and carcinomas (Supplementary Fig. 2A). These results demonstrate the power of multi-tissue comparisons and the application of the Gini index to efficiently identify tissue and disease-relevant genes. Whereas using only gene expression strength in a single tissue identified genes with general, housekeeping functions.

Many transcription factors (TFs) have restricted patterns of expression and are critical in developmental patterning and specification across many tissues[57]. Therefore, we asked if CF Gini genes are enriched for TFs. Amongst the CF Gini genes, we identified significant enrichment of human TFs annotated by ref. 57 (n = 71, 6.1-fold enrichment, Fisher exact p < 2.2 × 10⁻¹⁶). These included many known to be important in CF development, such as *ALX1, ALX4, DLX5, DLX6, MSX2, PAX3, PAX7*, and *TWIST1*[58]. In addition to transcription factors, signaling pathways are also well appreciated to be important for normal CF development. Consistent with this established biology, we found significant enrichment of genes belonging to families of signaling molecules known to be important in regulating CF morphogenesis, including *FGF8, FGF19*[59–61] and *WNT7A, WNT7B*, and *WNT8B*[62] (Supplementary Fig. 2B).

The genes with the highest Gini index for CF tissues are shown in Fig. 2c. The most highly expressed top-scoring gene, *COL2A1*, has been linked to Stickler syndrome, which includes CF abnormalities like cleft palate and midfacial hypoplasia[63,64]. While this approach successfully revealed genes known to drive CF disease, the unbiased nature of this analysis also allowed us to uncover candidate genes not previously associated with CF development. For instance, *HMGA2* is one of the most specifically expressed genes in our CF tissues. This gene has been linked in OMIM[5] to Silver-Russell syndrome (SRS) and is associated with a pygmy mouse phenotype[65], but is relatively underappreciated in both SRS specifically[66] and CF development generally[67]. Another top-scoring gene in our analysis, *IGF2BP1*, encodes an RNA-binding protein (RBP) that has not been linked to any human diseases in OMIM[5] but results in specific cartilage loss in nasal bones and mandible in

mouse[68]. RBPs, as a class, have been linked to a handful of syndromes with CF phenotypes[69], suggesting that *IGF2BP1* may also play a key role in CF development in humans.

To further determine if these CF expression biases reflected tissue and developmentally relevant differences, we asked whether such trends were also observed in mouse. To address this, we retrieved gene expression data uniformly processed by recount3[70] from developing mouse craniofacial prominences along with 20 other tissues ranging from embryonic day 10.5 (E10.5) to postnatal week 4, totaling 500 samples (Supplementary Data 1). Based on PCA, we found these data largely demonstrated expected differences in global gene expression that reflected their developmental origin (i.e., mesoderm, ectoderm, and endoderm), similar to our findings in the human datasets (Supplementary Fig. 2C). When we compared gene expression of orthologous genes across human and mouse craniofacial samples, we found that the second principal component oriented samples in a clear developmental trajectory (Supplementary Fig. 2D).

While it is difficult to directly compare developmental stages across species with dramatically different in utero development times, calculation of sample distances based on the expression of all one-to-one orthologous genes indicated CS13 samples were most similar to E10.5 while CS17 samples were most similar to E12.5 (Supplementary Fig. 2E). When we performed the same Gini analysis as above for these mouse tissues, we found similar enrichments of coherent biological functions amongst tissue-biased genes (Supplementary Fig. 2F). For instance, heart biased genes were enriched for functions related to muscle cell differentiation and sarcomere organization while craniofacial-biased genes were enriched for skeletal development and olfactory receptor activity.

Of the 239 human craniofacial Gini genes identified above, 154 had clear one-to-one orthology in mouse. For the remaining 5321 Gini genes, 2571 had clear orthology in mouse. Of the CF Gini genes with clear orthology, we found significant enrichment of genes with similar biases toward craniofacial samples in mouse ($n = 27$, Fisher exact $p = 5.663 \times 10^{-13}$, odds ratio = 7.58). These genes were enriched for craniofacial-relevant disease ontologies, as in the human findings above. Given the highly conserved expression during craniofacial development, we wondered if these genes are likely to physically interact or otherwise directly regulate one another. When we interrogated these genes, we found that these genes were very strongly enriched for both predicted and experimentally determined protein–protein interactions suggesting direct physical regulation amongst proteins encoded by these genes in both humans and mouse (Fig. 2d).

These findings suggested our multi-tissue and multi-species Gini analysis captured many known CF disease genes, revealed a conserved regulatory network at both expression and protein interaction levels, and implicated many novel genes that are biased in expression toward craniofacial development, warranting further inspection. Moreover, this analysis identified thousands of genes with restricted expression across tissue types, which are likely important to many different fields.

### Integration of chromatin states and gene expression from multiple tissues reveal trends in regulatory architecture for tissue-specific genes

Given the enrichment of genes with known roles in craniofacial development and significant numbers of transcription factors in our Gini analysis, we asked if tissue-specific regulatory sequences might play a role in these restricted expression patterns. Noncoding regulatory regions such as enhancers are known to act on genes in a spatiotemporal manner[71] and have been shown to be enriched for disease-associated variants[72,73]. In previous work, we have shown that enhancers active in human craniofacial and heart development are enriched for variants associated with risk for orofacial clefting and atrial fibrillation, respectively, suggesting they act on tissue-specific developmental genes[25,39].

To determine if CF Gini genes are likely regulated by CF-specific enhancers, we leveraged epigenetic atlases from embryonic development along with fetal and adult tissues in Roadmap Epigenome[35]. Using the "strong" enhancer states identified from the 25 ChromHMM analysis[74], we established a list of 10,693 putative CF-specific strong enhancer segments (CFSEs) by comparing them to all other available tissues in Roadmap Epigenome and embryonic heart (Supplementary Data 3). Using GREAT[75] to identify CFSE gene targets, we found that as the number of CFSEs predicted to target a gene increased, the bias of that gene's expression toward CF tissue increased (Fig. 3a). These results suggest that CFSEs drive tissue-biased gene expression and many are likely to be directly targeting these genes in vivo. We found that of all Gini genes, the CF Gini genes had the largest enrichment of CFSE target genes, followed by fetal brain and embryonic heart, confirming the global gene expression trends between tissues (Fig. 3a). Given this robust predicted targeting between CFSEs and CF Gini genes, we asked where tissue-specific enhancers tend to be physically located relative to their targets. To address this question, we assigned each CFSE to the closest CF Gini gene in cis. Compared to a random permutation control, we detected significant enrichment of CFSE-CF Gini gene pairs at distances up to 400 kb, with a median distance of 180 kb (Fig. 3b and Supplementary Fig. 3A, B). Approximately 20% of these CFSEs were located within 50 kb of a CF Gini gene. Enhancers have been demonstrated to form long-distance interactions with their target genes, with some of the most relevant and specific interactions occurring over distances greater than 1 Mb in developing tissues[76,77]. While such individual examples are striking for both the distances and specificity of effects, extremely long-range interactions (>500 kb) seem to be relatively rare based on these enhancer-gene pairings and do not rise above the background expectation.

The results above indicated that tissue-specific, active enhancer states play strong roles in the specificity of expression. However, repressive chromatin modifications and the complexes that deposit those marks have been associated with developmental abnormalities. Specifically, components of the Polycomb Repressive Complex 2 (PRC2), particularly *EZH2* that catalyzes H3K27me3 deposition, are essential for normal mouse development and have been implicated in neural crest-derived tissue specification as a result of global gene derepression[78–82]. The combination of H3K27me3 and H3K4me3 marks is strongly associated with bivalent chromatin state segmentations across multiple tissues. Bivalent domains are thought to be regions within the genome that facilitate temporal expediency during development but also identify genes with restricted or gradients of expression within a tissue[35,83–85]. Our previous work identified 957 genes with bivalent promoters exclusively in embryonic craniofacial tissue[25]. Many of these were DNA-binding factors that are known to have restricted patterns of gene expression across the mouse embryo, including the adjacent transcription factors *Dlx5* and *Dlx6*[86–88], suggesting a relationship between bivalent status and specificity of expression. When we examined all CF-specific bivalent genes, we did not find a correlation with tissue-specificity of gene expression (Supplementary Fig. 3C). However, bivalent genes classified as DNA-binding factors were significantly enriched for embryonic CF Gini genes. This trend was also observed for all bivalent promoters conserved between mouse and human-developing CF tissues (Fig. 3c).

Notably, we found that DNA-binding factors with bivalent promoter status during human CF development showed the strongest trends in both the absolute number of assigned CFSEs and the greatest overall specificity of expression. For example, the nuclear receptor *NR2F1* gene, which has been shown to be a key regulator of neural crest specification[89], is potentially targeted by 16 CFSEs. Other well-known CF TFs like *MSX2* and *ALX4* are targeted by ten or more CFSEs[90–93] (Supplementary Fig. 4).

This analysis also identified several genes that cause syndromes that include variable dysmorphic facial features but are not well

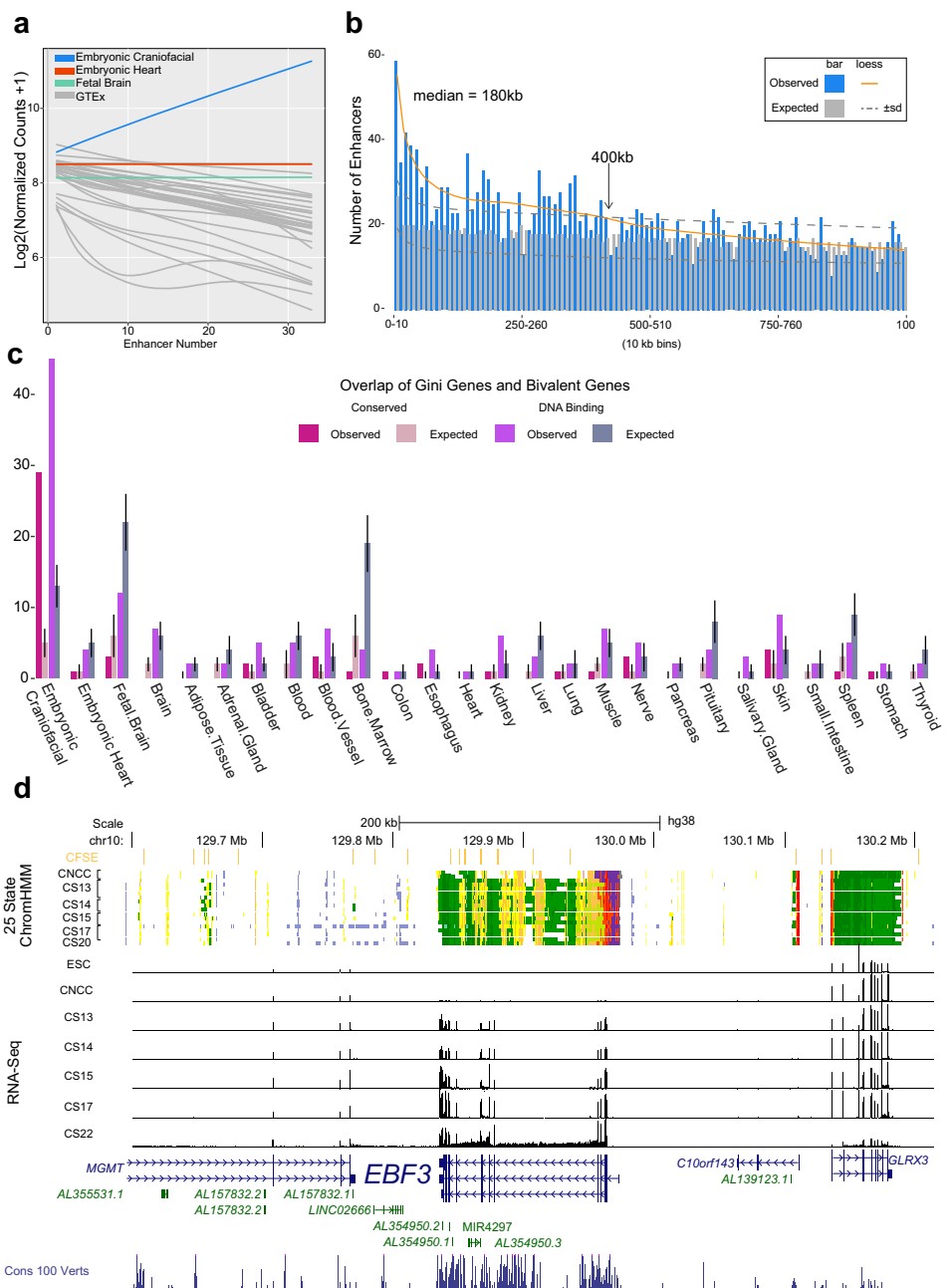

**Fig. 3 | Strong craniofacial-specific enhancers target Gini genes. a** Expression values versus CFSEs number modeled by linear loess smoothing for embryonic craniofacial (blue), embryonic heart (red), fetal brain (green), and all tissues in GTEx (gray). **b** Histogram plot of distance for closest craniofacial (blue bars) Gini gene to CFSEs compared to a randomly picked Gini gene from all other tissues (gray bars). The median and standard deviation were calculated from 1000 iterations. Each line is an interpolation for craniofacial Gini genes (orange) and for the maximum and minimum random Gini genes (dark gray dashed). **c** Gini gene enrichment for genes with bivalent promoters in craniofacial tissue. The pink bars are bivalent genes conserved between human and mouse, and the purple bars are bivalent regions

spanning DNA-binding genes. The bars for the expected overlap represent the median and lines represent the standard deviation calculated from 1000 iterations of randomly selected genes from the gini analysis (*n* = 33,073). **d** Genome browser shot of *EBF3* locus showing 25-state ChromHMM segmentation from culture model of CNCCs, and CS13 through CS20 of primary human craniofacial tissue. *EBF3* has a bivalent promoter in these samples (dark purple). Above the segmentation tracks is the active and novel CFSE track shown in orange, which includes strong enhancer ChromHMM states 13–15. Below are RNA-seq bigwig signal tracks for human embryonic stem cells (ESC), CNCCs, and CS13 through CS22.

appreciated in human CF disease, including *BCL11A*, *EBF3*, and *HMGA2* (Supplementary Data 3 and Supplementary Fig. 5). *BCL11A* has been recently implicated in Dias–Logan syndrome that is primarily known for intellectual developmental disability with hereditary persistence of fetal hemoglobin, but patients have been reported to have multiple dysmorphic features including microcephaly[94]. There are 12 CFSEs within 400 kb of *BCL11A* and up to 20 in the extended 2.3 Mb non-coding region flanking this gene (Supplementary Fig. 5). The potential

involvement of *HMGA2* in Silver-Russell syndrome has been discussed above. It is putatively targeted by 11 CFSE within 400 kb (Supplementary Fig. 5). In the case of *EBF3*, it has been linked to a neurodevelopmental syndrome that includes variable dysmorphic facial features (HADDS; MIM: 617330). There are 19 CFSEs within 400 kb of the bivalent TSS region and we observed a CF-specific pattern of expression compared to surrounding genes (Fig. 3d). Additionally, we observed that the 1.5 Mb window containing *EBF3, MGMT*, and *MKI67*

had the highest density of CFSEs genome-wide and could be an underappreciated locus for craniofacial development and associated abnormalities. Overall, these results demonstrate a clear relationship with the genomic organization of tissue-specific enhancers relative to genes that have high specificity of expression during development, particularly those that maintain a bivalent state during organogenesis.

## Differential expression during craniofacial development identifies temporal-specific biological processes

Our analysis thus far has demonstrated the value of comparing across multiple tissues for observing global trends of gene expression and identification of genes with tissue-specific patterns of expression. However, these analyses assume each tissue is at a one-timepoint and thus do not capture gene dynamics throughout development. Craniofacial development is a morphologically complex process which is likely to be reflected in its gene expression, thus we sought to understand how gene expression patterns change throughout early

development in craniofacial tissues. We first performed differential expression analysis in a pairwise fashion across all polyA-captured RNA from primary tissue timepoints. We found a total of 3677 genes significantly differentially expressed. The largest number of differentially expressed genes (DEGs) were found in comparisons between the earliest and latest stages (CS13 and CS17, $n = 2733$), consistent with the progression of morphogenesis of a structure into multiple distinct tissue types (Supplementary Fig. 6A–C and Supplementary Data 4).

When we compared the expression of these DEGs across all of our gene expression data, we found that hierarchical clustering separated samples into two main groups that we denote as Early (CNCC, CS13, and CS14) and Late (CS15, CS17, and CS22) (Fig. 4a). This grouping of samples indicated the culture model of CNCC utilized in this study does indeed reflect the primary tissue when neural crest cells are thought to be populating the craniofacial structures. This is in contrast to an alternative cranial neural crest cell culture model[42] which seems to have very distinct expression profiles compared to the CNCC data

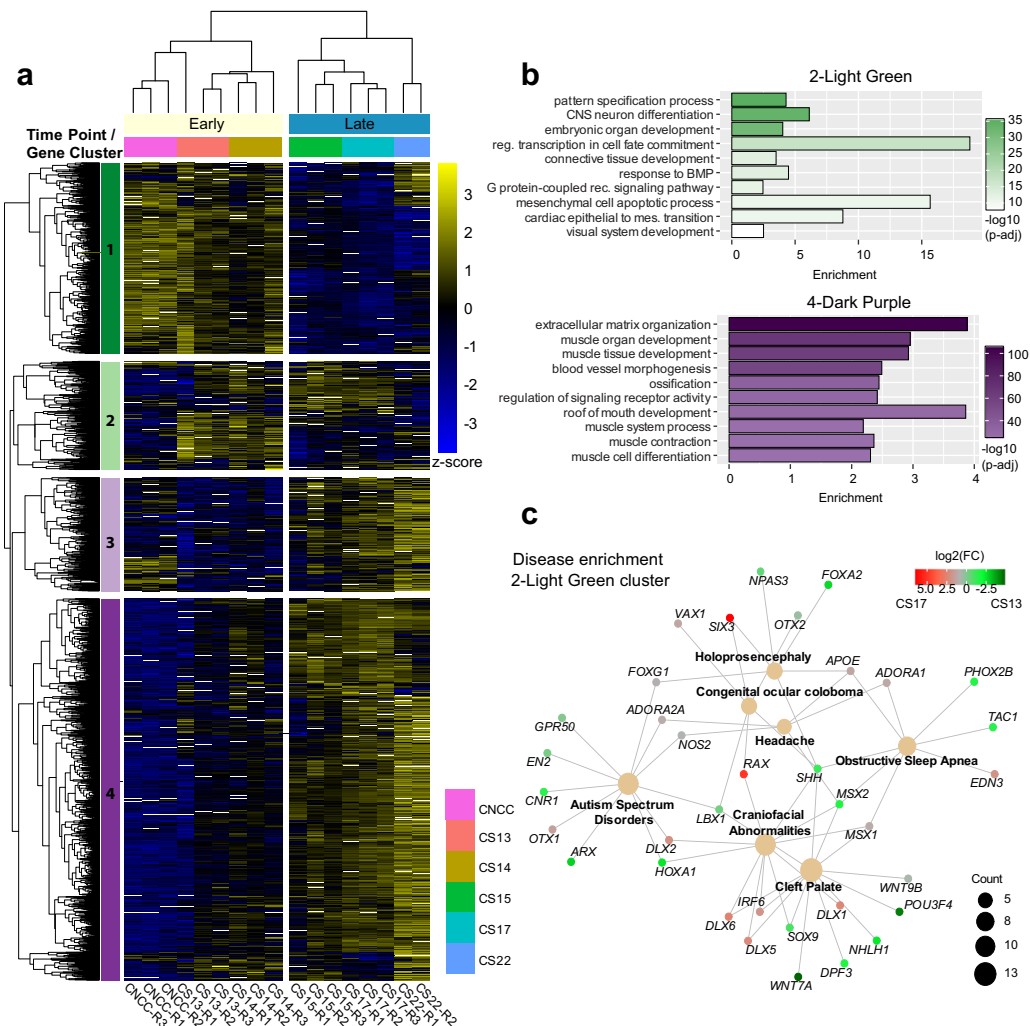

**Fig. 4 | Differential expression during craniofacial development. a** Heatmap of differentially expressed genes (log₂ fold change > abs(1), Benjamini–Hochberg *p*-adj <0.05) across all samples in our developmental series. *Z*-scores for normalized gene counts ranging from high (yellow) to low (blue) expression. The dendrogram on the top is the hierarchical clustering of all samples. Samples were separated into two main clusters, Early (CNCC, CS13, and CS14) and Late (CS15, CS17, and CS22). The dendrogram on the left is hierarchical clustering of gene counts. The genes were assigned groups identified by color and number (1-dark green, 2-light green, 3-light purple, and 4-dark purple) by cutting the dendrogram at a height that would result in four clusters. The genes in the green clusters are those primarily expressed

in the early and intermediate stages of the series. While the purple cluster genes tend to be active in the late stages. **b** Gene ontology of a select subset of the most significant terms identified by clusterProfiler[146] for 2-light green and 4-dark purple clusters. The genes of the 2-light green cluster enrich for embryonic patterning and terms concerning the mesenchyme. The 4-dark purple cluster includes muscle developmental functions with ossification and blood vessel morphogenesis. **c** Network plot of selected significant disease ontology terms from genes of the 2-light green cluster. Count refers to the number of genes within a category and the colorbar shows log₂ fold change as calculated by DESeq2[141] between CS13 and CS17 samples which color the gene nodes.

and primary tissue gene expression data generated in this study (Supplementary Fig. 6D).

We identified four major clusters of genes which had obvious expression trends during the developmental trajectory (Supplementary Data 4). The upper cluster (1-Dark Green) showed strong expression in CNCC and CS13 samples and progressive dampening of expression over developmental time. The genes in the upper mid cluster (2-Light Green) are largely absent in CNCC and CS22 but have dynamic expression between early and late stages (Fig. 4a). These two clusters were enriched for genes involved in patterning, morphogenesis, and cell fate specification (Fig. 4b and Supplementary Fig. 6E). Disease enrichment analysis of the light green cluster revealed strong enrichment for CF abnormalities including cleft lip ± cleft palate (Fig. 4c). Genes in the lower mid cluster (3-Light Purple) were expressed in CNCC but largely absent from the rest of the early tissue samples. They are expressed again at CS15 and generally increase across the rest of the developmental trajectory (Fig. 4a and Supplementary Fig. 6E). Genes in the bottom cluster (4-Dark Purple) are mostly not expressed in the early stages but progressively increase in expression over the duration of the developmental trajectory. These two clusters were enriched for GO terms involved with cell–cell adhesion, muscle development, ossification, and upper palate development (Fig. 4b and Supplementary Fig. 6E). These DEGs included all members of the RUNX family of TFs, many members of the myogenic TF family, and all members of the PITX family of TFs.

Several genes identified to be involved in establishing neural crest identity[95] were found in all but one of the clusters, with most residing in dark purple. Interestingly, *PAX7* and *RXRG* exhibit oscillating trajectories, which varied from the overall trajectory of the dark green cluster (Supplementary Fig. 7A, B). This may be due to multiple waves of neural crest populating the human CF tissue, as it is thought to happen in trunk sensory neurogenesis[96]. These data suggest that the embryonic period of development, particularly the 4 to 8 post-conception week period in humans, has distinct programs that are employed sequentially as morphogenesis and organogenesis of the face and skull progress.

The classes of genes linked to disorders with CF abnormalities are exceptionally varied. They include TFs, cell–cell signaling, RNA-binding, and protein metabolism[97,98]. We asked if certain classes of genes would be biased toward dynamic expression in our CF tissues. We found that signaling and TF classes were more prominent in our differentially expressed genes than expected (Supplementary Fig. 7C) while RBPs were depleted. Many RBPs are known to be ubiquitously expressed and we show that they have generally low Gini scores in our multi-tissue analysis (Supplementary Fig. 7D). It is therefore likely that their function is not directly regulated by transcription, but instead may be important for post-transcriptional processing of genes that are either specifically expressed or expressed in cell types like migrating neural crest that are particularly sensitive to perturbation[99].

Based on our above multi-tissue analysis, specifically expressed genes and TFs with bivalent chromatin status are indicated to play important roles in CF development. We found that CF Gini genes were most enriched in the early green clusters (Fisher exact $p < 2.2 \times 10^{-16}$). Bivalent transcription factors showed similar trends in enrichment across clusters, with the most significant enrichment in the light green and dark purple clusters (Fisher exact $p < 2.2 \times 10^{-16}$), suggesting that throughout development, there are several narrow windows of time when craniofacial-specific biological processes are occurring.

When we analyzed CS13 chromatin states at the promoters of genes in each cluster, we found consistent trends in activating histone modifications, particularly H3K4me3 levels of the dark green cluster that are more strongly expressed in the early time period (Supplementary Fig. 8). Additionally when we inspected differential enhancer utilization across development we found enhancers of DEGs showed similar trajectories of H3K27ac levels (Supplementary Fig. 9). Overall

the patterns of expression that were revealed in the differential expression analysis were largely confirmed by chromatin state both at promoters and distal enhancers suggesting direct regulation of genes by changes in histone modifications at these classes of regulatory sequences.

## Coexpression networks reveal groups of genes with distinct roles in craniofacial development

The differential expression analyses above revealed clear dynamics in gene expression across a subset of genes. However, genes not rising to stringent pairwise differential expression criteria are not considered, which potentially misses information about other important gene relationship patterns such as coexpression. Gene coexpression during development has been shown to reveal modules of genes that have similar patterns of regulation across time and even identify cell-type specific enrichments from bulk data[39,100,101]. Coexpression network analysis has been successfully applied to a variety of contexts ranging from the developing brain, particularly for unraveling the complex genetic architecture of autism spectrum disorder (ASD), to the developing tooth to suggest odontogenic loci[102–104]. We wondered if the same trends might exist in our CF expression data and could be used to reveal putative disease gene candidates as demonstrated in ASD[105]. To attempt to uncover such genes, we applied weighted gene coexpression network analysis (WGCNA)[106] using similar approaches as those employed in ASD studies[102]. After normalization and filtering for low expression, 26,626 genes were assigned to 29 modules ranging in size from ~150 to over 4000 coexpressed genes. We hypothesized that there would be modules unique or important to CF development, and to help identify them, we performed GO enrichment (Fig. 5a). The GO analysis revealed that the modules were biologically distinct from one another (Supplementary Data 5). The decreasing modules captured by WGCNA were enriched for many generic cellular processes such as cell cycling (orange), tRNA processing (green), splicing (dark green), and protein trafficking (dark orange). Amongst the increasing modules, more tissue-specific enrichments were observed, including muscle development (dark magenta), epithelial differentiation (light yellow), and, most notably embryonic patterning and morphogenesis (black).

We then calculated representative gene expression for each module, or module eigengene expression, and inter-module eigengenes correlation and relatedness. Modules located near one another in multidimensional scaling space (MDS) (Fig. 5a) revealed three trajectories of expression across development: increasing, decreasing, or dynamic (Fig. 5b and Supplementary Fig. 10). To get a finer characterization of these modules, we determined enrichment from twelve gene lists including craniofacial related abnormalities (Supplementary Data 2), specificity of expression, assignment of CFSE to target genes, clusters of differential expression, and measures of resistance to deleterious mutations (Fig. 5c). We found that two modules had enrichment of known CF disease genes and four modules were enriched for CF Gini genes. When we analyzed modules for potential regulation by CFSEs, we found eight modules were enriched and were heavily biased toward modules with dynamic patterns of gene expression. Three modules, light green, black, and dark slate blue had enrichment for both CF Gini and CFSE, further establishing a connection between enhancer activity and gene expression. The coexpression networks also captured relevant gene dynamics since many of the modules showed enrichment for the DEGs from Fig. 4.

We next turned to exome and genome sequences collected by gnomAD[107], which has developed a score based on observed versus expected numbers of loss-of-function mutations across these data (loss-of-function observed/expected upper bound fraction or LOEUF). Genes identified in the upper deciles of this scoring metric have been previously predicted to most likely cause disease in humans. We found that 8 WGCNA modules had significant enrichment for the upper

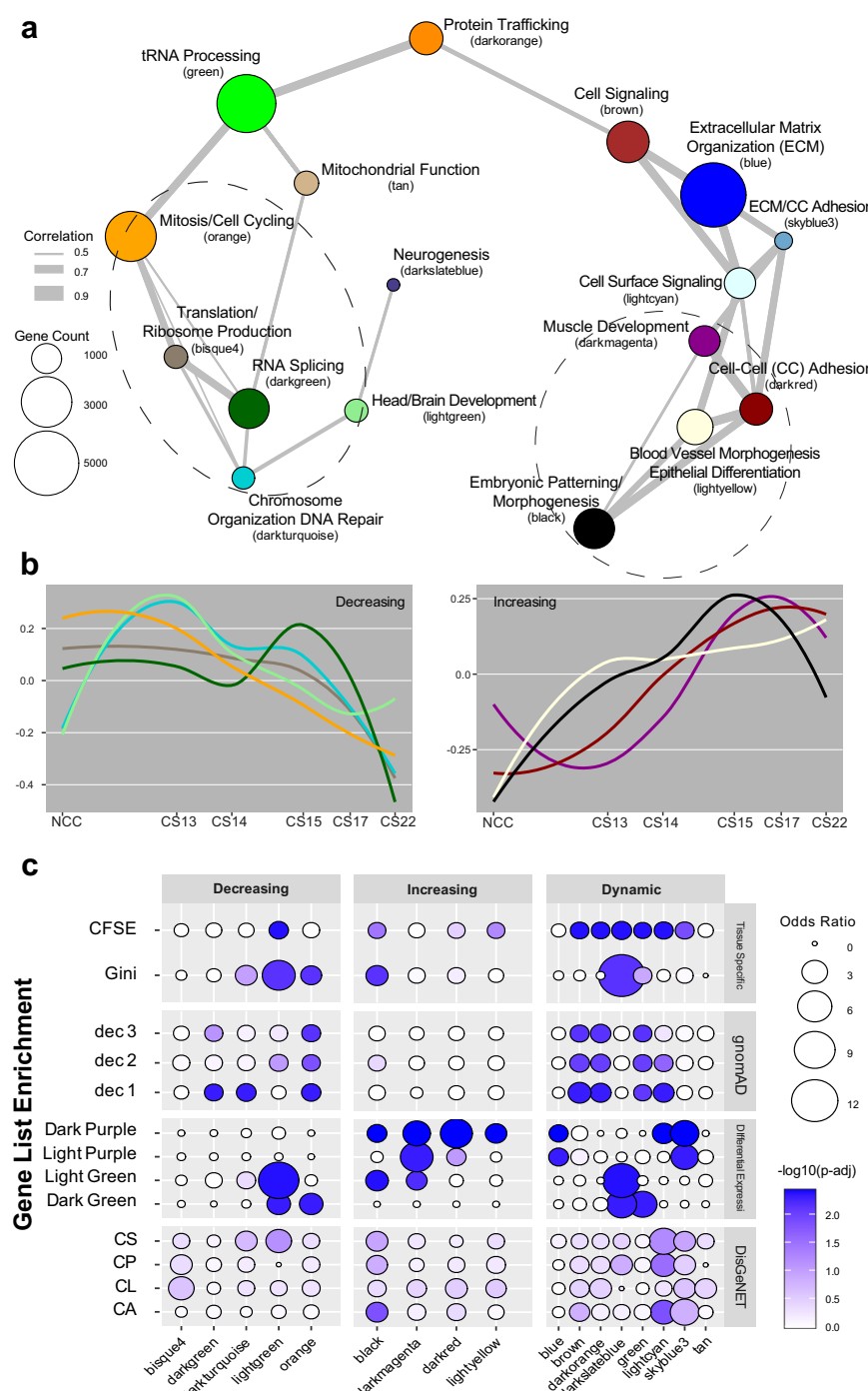

**Fig. 5 | Developmental transcriptome organized into modules of coexpressed genes characterized by curated gene sets. a** Network plot of gene modules identified by WGCNA. A Pearson correlation of the module eigenvectors was calculated for the edges where positive correlations of ≥0.5 were included. The location of each module is determined by multidimensional scaling (MDS) of the module eigengene vectors. Modules are color-coded based on names assigned by WGCNA. The size of the dots indicates the number of genes in each module. Each module is labeled based on the most significant biological process category, gene ontology enrichment, determined by DAVID; however, this label is not always all-encompassing. See Supplementary Data 5 for an exhaustive list. Modules were grouped based on proximity and eigengene trajectory over developmental time. **b** Trajectories of expression based on eigengene vectors reported by WGCNA for

each module across the developmental time series. The modules are assigned to one of three groups based on if it is generally decreasing, increasing, or oscillating over developmental time. See Supplementary Fig. 10 for all module trajectories. **c** Dot plots of gene list enrichment within the WGCNA modules. Gene lists (rows) are divided into four groups based on their source. Dec 1 through dec 3 are genes from these deciles, as reported by gnomAD. Disease gene lists from DisGeNET[56], CS craniosynostosis, CP cleft palate, CL cleft lip, CA craniofacial abnormalities. The rest of the lists are generated from data in this study. The dot color is determined by the Benjamini–Hochberg-adjusted permutation *p* value (see Methods), with darker indicating higher significance. Modules (columns) are grouped by their trajectory assignment from **b**.

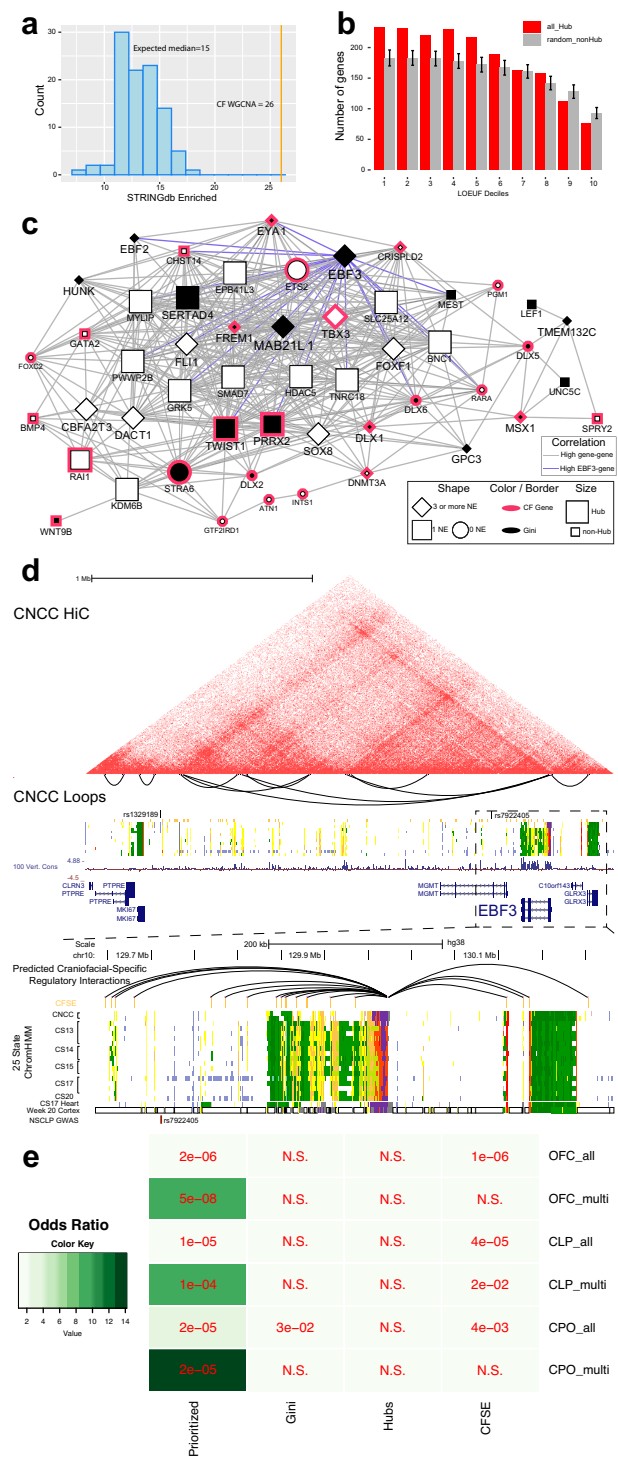

**Fig. 6 | WGCNA network hub-gene enrichment of variant-restricted genes uncover disease risk of EBF3 for craniofacial development. a** Histogram of the number of gene-scrambled modules that have protein–protein interaction (ppi) enrichment at a Bonferroni-adjusted one-sided *p* value <0.05. The vertical orange line marks the number of modules that have significant ppi in the observed WGCNA network (permutation *p* value = 0.01). **b** Histogram of (LOEUF) deciles of hub genes and randomly selected non-hub genes from all modules in the WGCNA network. Deciles range from decile 1 (d1), which represents the most constrained genes, to d10, genes that are the most tolerant to putative loss-of-function (pLoF) variation. The bars for the expected overlap represent the median and lines represent the standard deviation calculated from 1000 iterations of randomly selected genes available from gnomAD (*n* = 17,277). **c** Network showing some of the top-scoring genes (Supplementary Data 6) in the black module, along with known disease genes. The edges represent the Pearson correlation (>0.9) between genes. Blue edges connect genes that are highly correlated to EBF3. Genes are represented by nodes which are characterized by shape, fill color, border color, and size. A black fill indicates the gene has craniofacial-specific expression (Gini ≥0.5). Known disease genes are marked with a Fuschia border. If the gene has been assigned three or more CFSEs it has a diamond shape, one CFSE is square, and 0 are circles. A node with a large size is a hub gene. Genes which have several of these criteria are displayed with larger and bold text. **d** CNCC Hi-C interaction plot of the *EBF3* locus, data from[121]. Below is a zoomed-in genome browser shot of *EBF3* locus showing 25-state ChromHMM segmentation from a culture model of CNCCs, primary embryonic CF tissue, embryonic heart and fetal cortex, and CS13 through CS20 of primary human craniofacial tissue. Two orofacial clefting GWAS SNPs reside in and near the TAD boundary. **e** Significance and fold enrichments of overlaps of genes from our prioritization strategy (Prioritized), CF Gini genes (Gini), WGCNA hubs (Hubs), and genes targeted by CFSEs (CFSE) with genes identified with at least one DNM (All notation) or multiple DNMs (Multi notation) from orofacial clefting trios.

## Regulatory hubs in coexpression networks reveal putative driver genes for the developing face

Given the robust enrichment of multiple aspects of CF biology across multiple data types in the modules of our WGCNA, we set out to determine if these modules could reveal genes involved in CF development or cause disease that had been previously inaccessible. Within coexpression networks, potential driver genes or "hub genes" are those that have correlations with many other genes (high connectivity). Previous analysis of coexpression networks of early brain and heart development showed hub genes are enriched for genes with strong evidence of involvement in ASD and congenital heart disease (CHD) risk, respectively[39,102,103]. We thus hypothesized that hub genes, in particular, would be enriched for known and candidate disease genes.

We first tested all hub genes against the gnomAD database and observed significant enrichment across upper LOEUF deciles and significant depletion of lower LOEUF deciles (Fig. 6b). When we examined the hub genes of each module independently, we identified seven modules whose hub genes were enriched across the two upper LOEUF deciles. These included all of the modules identified above (Fig. 5c), with the exception of the light green module. Of these seven, the dark turquoise, dark red, and black modules were enriched across the top four deciles and had the highest fold enrichments over background permutations. When we combined these enrichments with the previously described annotation analyses and gene ontology enrichments performed, we found the black module was singularly enriched for all these metrics.

Specifically, the black module was significantly enriched for genes linked to craniofacial diseases, genes involved in embryonic morphogenesis, CF Gini genes, genes targeted by CFSEs, genes constrained in healthy humans, and genes which are differentially expressed across the CF developmental trajectory. We identified a number of driver genes within the black module that are well-known to the CF development field, including *DLX5, DLX6, EYA1, MSX1, PRRX2, MSX1,* and *TWIST1*. Given this enrichment of known disease genes, we wondered

LOEUF deciles (Fig. 5c). Many of these same modules were also enriched for both known disease genes and CF Gini genes.

One implication of the coexpression of genes is that they may be more likely to interact at the protein level. For instance, genes that encode proteins that are members of the same complex or biological process would need to be expressed within similar points of development. So we asked how our coexpression networks might reflect protein–protein interactions (PPI). We found that, indeed, our WGCNA network was strongly enriched throughout for known and predicted protein–protein interactions (Fig. 6a). Overall, these findings strongly suggested that coexpression networks reflect the known biology active during craniofacial development.

whether other genes in this module might be strong disease candidates. Two of the most prominent such genes in the black module were *MAB21L1* and *EBF3* (Fig. 6c). These genes shared features of known CF disease genes in the black modules, but were more highly connected, targeted by highest numbers of CFSEs, specifically expressed in CF tissues, but not widely known to play roles in CF development or disease based on annotations in OMIM and DisGeNET. When we inspected the literature, we found that *MAB21L1* has recently been identified to be responsible for an ultra-rare disorder which notably involves ocular anomalies, facial dysmorphism, and microcephaly[108]. *Mab21l1* KO mice have ocular anomalies and lack of closure of the anterior fontanel[109]. Homozygous *Ebf3* mutant mice die perinatally but do not have overt CF defects[110]. Heterozygous *Ebf3* mutant mice have defects in olfactory neuron projection consistent with reported patterns of *Ebf3* expression being strongest in the olfactory epithelium[111,112]. Only a small number of human patients have been identified with rare de novo variants in *EBF3* but these patients consistently displayed intellectual disability, ataxia, and facial dysmorphologies[113–118].

We also find that an ortholog of *EBF3*, *EBF2*, is highly coexpressed (Fig. 6c) consistent with previous studies showing these genes overlap in expression and function[119,120] suggesting that this gene family may play an unappreciated role in human CF development. When we inspected the potential regulatory landscape of *EBF3* based on HiC[121] in the same CNCC system used for gene expression in this study, we observed a topologically associating domain (TAD) that extended from nearly 1.5 Mb from the *EBF3* promoter region (Fig. 6d). This TAD is the same window we identified above as enriched for CFSE segments and harbored two distinct loci linked to risk for orofacial clefting[122,123]. These two loci (rs1329189 and rs7922405) were previously labeled as being associated with *MKI67* and *MGMT*, respectively, based solely on proximity. However, *MKI67* has been used frequently as a marker of proliferating cells and *MGMT* is broadly expressed across human tissues with a Gini index of 0.23, very similar to *SOD1* in our above analysis. Thus, we nominate *EBF3* as the risk gene for orofacial clefting associated with these two loci.

Haploinsufficiency of *EBF3* was reported by multiple groups to cause hypotonia, ataxia, and delayed development syndrome, but orofacial clefting was not reported as a consistent feature[109–114]. A search of the exome sequencing clinical database at Baylor Genetics revealed 15 individuals whose phenotypes are likely to be explained by deleterious variants in *EBF3*. Three of these individuals had a cleft palate. The first individual was a previously published 13-year-old female with a submucosal cleft palate and a short uvula[113]. She carried a de novo likely pathogenic c.512 G > A, p.(G171D) [NM_001005463.3] variant in *EBF3*. The second was a six-year-old male with a cleft soft palate who carries a pathogenic de novo c.487 C > T, p.(R163W) [NM_001005463.3] variant in *EBF3*. He was born prematurely and had intrauterine growth restriction. Polyhydramnios was noted during pregnancy. His other phenotypes include delayed speech and language development, delayed motor milestones, hypotonia, flexion contracture, a positive Gower sign, strabismus, nasolacrimal duct obstruction, atrial septal defect, patent foramen ovale, obstructive sleep apnea, laryngomalacia, and macrocephaly. The third was a 5-year-old female with a cleft palate who carries a de novo likely pathogenic c.661 G > A, p.(V221M) [NM_001005463.3] variant in *EBF3*. She was also noted to have delayed speech, intellectual disability, hypotonia, epicanthal folds, micrognathia, camptodactyly, and hyperextensibility.

This suggests that the rate of cleft palate among individuals with putatively causative sequence variants in *EBF3* is approximately 20% (3/15; 95% confidence interval (CI) = 4.3–48.1%). Conservatively assuming that the rate of cleft palate is 1:500 in the general population[4], the rate observed in individuals with putatively causative *EBF3* variants is statistically higher (P < 0.0001; two-tailed Fisher exact test).

To further determine if haploinsufficiency *EBF3* is associated with an elevated occurrence of orofacial clefting, we examined the DECIPHER[124] database for individuals with copy number variants (CNVs) encompassing this gene. We identified 202 individuals with a deletion that affected *EBF3*. 14% (29/202) of these individuals were described as having a high palate, and 3.5% (7/202) were described as having a narrow palate, with six individuals having both a high and narrow palate. One individual was indicated to have a palatal fistula and 12 individuals were indicated to have cleft palate or cleft lip ± cleft palate. This represents a nearly 30-fold higher rate of orofacial clefting (6%, 12/202, 95% CI = 3.1–10.2%) compared to an incidence of orofacial clefting in the general population of the United States[4] (P < 0.0001; two-tailed Fisher exact test).

Given the findings above in the black module, we inspected all modules from the WGCNA that had the following traits: CF Gini genes, CFSE target genes, hub genes, and high LOEUF (decile ≤3) genes. There were 539 genes that fulfilled at least two of these criteria across our entire dataset (Supplementary Data 6, 7). To determine if this list might indeed be enriched for disease-causing genes, we first examined genes recently curated by CleftGeneDB[125] to have experimental evidence for orofacial clefting in humans and/or mice. We found a very significant overlap between the curated list and genes identified here that had been prioritized based solely on gene expression and enhancer dynamics in normally developing human craniofacial tissue ($n = 119$, Fisher exact $p$ value = $5.1 \times 10^{-91}$, odds ratio = 19.1, Supplementary Data 7).

We next turned to recently published genome sequencing data in isolated orofacial clefting trios[126]. This study reported only three genes (*TFAP2A*, *IRF6*, and *ZFHX4*) that had enough de novo mutations (DNMs), either protein-altering or loss-of-function, across probands to rise above conservative multiple $p$ value correction thresholds given the size of the cohort. However, there were significantly higher numbers of protein-altering DNMs in probands than expected by random chance. In this study, it was hypothesized that such genes might play a role in orofacial clefting risk, but to prove this, either much larger cohorts would be necessary, as has been seen in the study of ASD, or additional experimental corroboration would be needed. We hypothesized that our datasets and, specifically, the prioritization scheme above could provide additional support for the genes reported.

To test this hypothesis, we inspected data from these 756 trios along with an additional dataset of 435 trios with cleft palate (CP)[127]. We identified 875 genes with at least one DNM and 58 genes with multiple DNMs in probands across this cohort. These could be further divided into genes with at least one DNM in probands with cleft lip with or without cleft palate (CL/P) and those with cleft palate (CP) (572 and 375 genes with one DNM, 29 and 21 genes with multiple DNMs, respectively). We then interrogated these gene sets for significant overlaps with gene lists that we have assembled here, specifically CF Gini genes, CFSE targeted genes, hub genes, and prioritized genes. In contrast to findings in ASD, we did not find any significant overlaps between any of the DNM gene sets and hub genes for individual modules from WGCNA or hubs as a total. For CF Gini genes, we only found significant enrichment for genes with a DNM in CP. For genes putatively targeted by at least one CFSE, we found significant, but small enrichment for each category of all DNM genes, although the only category of multiple DNMs enriched was for CL/P. Surprisingly, we found significant overlaps with greatest fold enrichments with our prioritized genes and all the DNM gene sets. For each OFC grouping, the genes with multiple DNMs were more significantly enriched in our prioritized genes than those with a single DNM (Fig. 6e and Supplementary Data 7).

Amongst genes with multiple DNMs, our prioritization strategy identified all three genes reported by Bishop et al. (*IRF6*, *TFAP2A*, and *ZFHX4*)[126]. Other genes included collagen genes *COL2A1* and *COL11A1*, transcription factor *GRHL3*, cell adhesion and signal transduction associated gene *CTNND1*, and intraflagellar transport gene *IFT122*.

Together these results provide additional support for these genes as bona fide clefting genes. These results also suggest our genomics data-driven prioritization strategy is enriched in genes with elevated DNM rates in orofacial clefting and reveal gene expression alone is insufficient for the prioritization of genetic variants in this class of developmental defects. Moreover, the significant overlap with curated lists of human and mouse clefting genes suggests the rest of our prioritized genes, particularly those with a DNM observed in clefting probands, are fertile ground for identifying additional human disease-relevant loci.

## Single-nuclei RNA-sequencing of embryonic face

Our multi-layered analyses implicated *EBF3* as an important player in gene regulation during human craniofacial development and a risk factor for orofacial clefting. However, as detailed above, mice lacking *Ebf3* do not have any reported craniofacial abnormalities. In our human Gini analysis, we found that *EBF3* expression was biased toward craniofacial development (Gini = 0.65); however, in mouse *Ebf3* was more ubiquitously expressed (Gini = 0.11). Given these distinct differences in reported loss-of-function phenotypes and bulk gene expression patterns across species, we hypothesized that this gene and its paralog *EBF2* might[104] be expressed in different locations or cell types in human craniofacial development compared to mouse.

While in situ hybridization for *EBF3* might be a straightforward way to address this issue in mouse embryonic tissues, this is a much more challenging proposition in human embryonic samples. The samples we have used above and to which we currently have access are all dissected from the rest of the embryo and flash-frozen. Thus, the samples cannot be embedded for sectioning easily without thawing, and many of the structural landmarks for the orientation of the tissue are absent. We have shown in multiple studies that despite these orientation-based issues, tissue cryopreserved in this way is useful for high throughput sequencing experiments which do not require orientation information. Therefore, we looked to address potential species-specific differences in *EBF2/3* expression using single-cell expression assays. We focused on the human timepoint with the highest median *EBF3* expression, CS17, and the most similar timepoint in mouse craniofacial development based on our bulk expression, E12.5 (Supplementary Fig. 2E). We performed single-nuclei RNA-seq (snRNA-Seq) on craniofacial prominences from three CS17 embryos, and also performed snRNA-Seq on craniofacial prominences from three separate pools of tissue from E12.5 mouse embryos.

To identify conserved cell types and their potential differences across species, we co-projected these samples using only genes with 1:1 orthologs in human and mouse (Supplementary Fig. 11). With these ~16,000 genes, we used GO enrichment and canonical marker genes to identify seven major cell populations (Mesenchyme, ectoderm, endoderm, muscle progenitor cells, red blood cells, other blood cells, and neural progenitor cells), consistent with the major resident cell types described in developing mouse CF tissue[128] (Fig. 7a and Supplementary Data 8). The relative proportions of each cell type were remarkably similar across species despite potential heterochrony (Supplementary Fig. 11); we also observed a strong concordance between analogous cell types across species, except neural progenitor cells. This was expected, as we have noted the large biases in the inability to finely dissect primary human CF tissue compared to the tightly controlled collection of mouse CF tissue. Therefore, much of the primary human tissue was expected to contain portions of the forebrain located adjacent to the face at this stage.

As discussed above, WGCNA is able to identify coexpressed genes primarily expressed in specific cell types even from bulk data[39,103,104,129]. To identify these gene signatures in our modules, we considered hub genes from each WGCNA module as a collective group and calculated a "module score" based on the combinatorial activity of these groups of genes in each cell. These gene module enrichment scores were indeed enriched for specific cell types (Fig. 7d and Supplementary Fig. 12). For example, the hub genes of the black module showed enrichment for the mesenchyme and endoderm cell populations, while the light cyan and light green hub genes are specifically expressed in the mesenchyme. Interestingly, when we interrogated our epithelial and mesenchymal subclusterings, we were able to observe an enrichment of WGCNA modules which did not appear in the bulk analysis. For example, when we look at all cell types, the dark slate blue module appears to be primarily enriched in the neural progenitor cell pool; however, when we look at the epithelial subclustering, there is a clear enrichment of hub genes from this module in epithelial cluster 4. We observe a similar finding in the mesenchyme for the light yellow module.

Given the enrichment of expression of genes from the black module, which contains *EBF2* and *EBF3*, in the mesenchyme and endoderm cell populations, we wondered whether these two genes would be significantly expressed in these cell types and potentially biased in expression across species. We observed elevated expression of *EBF2* in the mesenchyme and *EBF3* in the mesenchyme and endothelium of both mouse and human samples (Fig. 8a). When we inspected coexpression of these two genes, we found a striking bias toward human cells to jointly express them. This was particularly apparent in the mesenchyme (Fig. 8b). These results in the mesenchyme were of particular interest due to this cell type's relevance to craniofacial phenotypes. Therefore, we examined the mesenchymal subclustering for expression of *EBF2/EBF3* across species in these subpopulations. For clusters with median expression greater than 1, we observed significantly human-biased expression for both genes in mesenchyme subclusters 0 and 1 (Fig. 8c). In contrast, we did not find significant mouse-biased coexpression of these two genes in any subcluster. *Ebf3* was biased toward mouse subclusters 4 and 5, while *Ebf2* was mouse biased in subcluster 6. Of the human-biased subclusters for both of these genes, subcluster 0 and 1, key marker genes that differentially identify subclusters 0 and 1 were *RUNX2* and *RUNX1*, respectively (Fig. 7b). In mouse at E13.5, these two genes show distinct patterns of expression. *Runx1* was biased toward the cartilage and osteogenic regions of the upper palate and maxilla, while *Runx2* expression was biased toward the osteogenic mesenchyme in the mandible[130]. This suggests that mesenchyme subcluster 1 is likely to be derived from the upper palate. This is consistent with our genetic findings above of *EBF3* being a risk factor for orofacial clefting in humans.

## Discussion

Craniofacial malformations caused by genetic perturbations in craniofacial development are one of the largest classes of human birth defects; however, the gene expression patterns active in these tissues and structures have until now been largely uncharacterized. Here we have described the most comprehensive analysis to date of gene expression from rare human embryonic craniofacial samples. We have performed large-scale comparisons of gene expression from other tissues across the body and developmental stages. In order to prioritize and identify driver genes and regulatory networks, we have subsequently integrated these data with other types of functional genomics data, including ChIP-seq, Hi-C, and single-cell RNA-Seq from embryonic stages of CF development.

We demonstrated that the developing CF transcriptome is its own distinct tissue by performing comparisons of our CF samples to thousands of samples across dozens of tissues collected by the GTEx consortium as well as developing heart and brain. The employment of an unbiased measure of specificity, the Gini index, has identified many genes that are expressed exclusively and most strongly in these novel samples. Some craniofacial abnormalities are caused by defects in genes that are widely transcribed across many tissues yet manifest as relatively specific phenotypes. For example, mandibulofacial

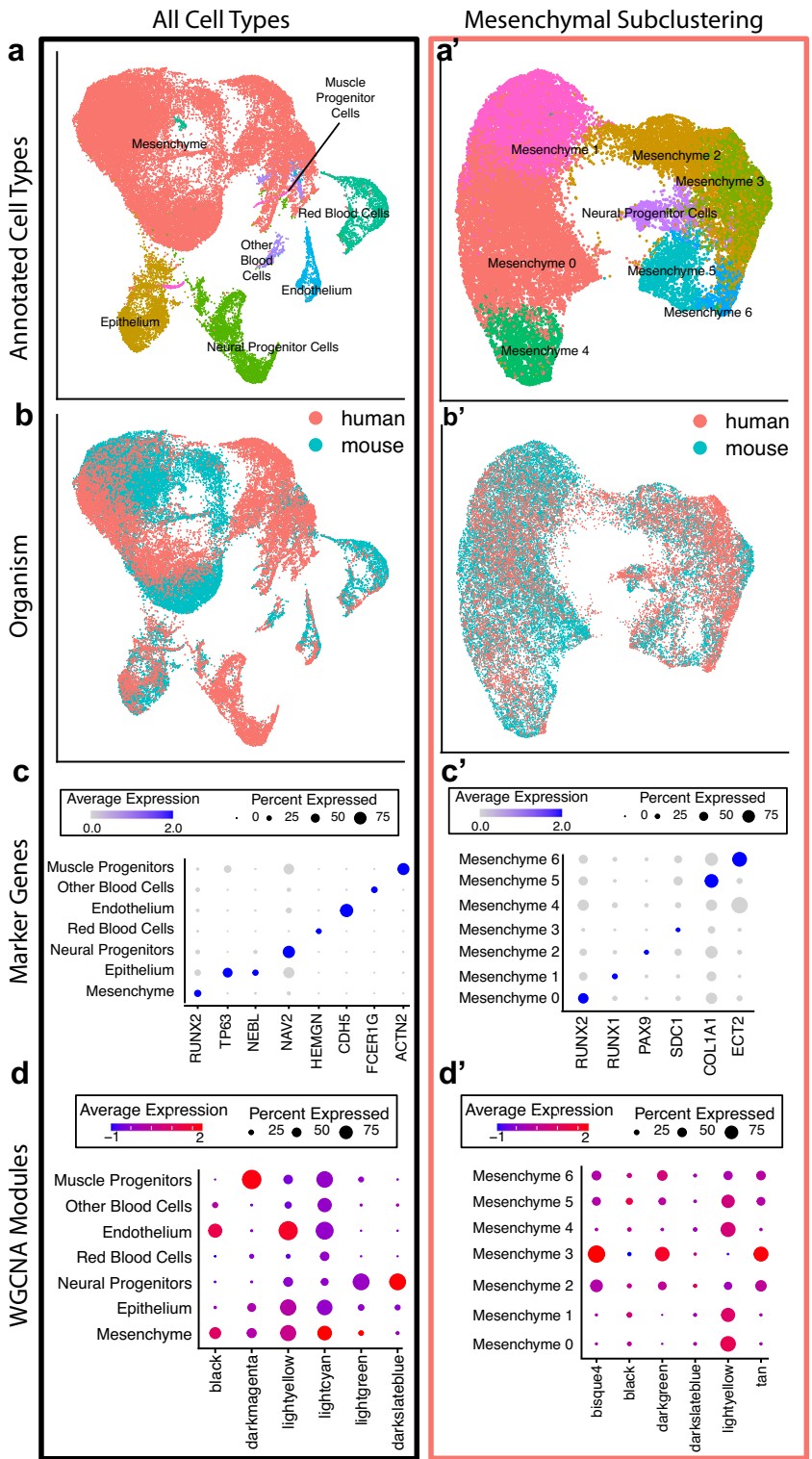

**Fig. 7 | Single-nuclei RNA-sequencing of developing mammalian face demonstrates conservation of cell types across species. a**, **a**′ Co-projected UMAPs of CS17 and E12.5 craniofacial tissues of (**a**), all cell types (**a**′), cells from "Mesenchyme" cluster in (**a**). Combined UMAPs were generated by subsetting matrices to only include genes with 1:1 orthologs across species. Generated via Seurat[145]. **b**, **b**′ Co-projected UMAPs as in **a**, **a**′, overlaid with the species identity of each cell (red = human, blue = mouse). **c**, **c**′ Dot plots of gene expression for canonical cell type marker genes in each of the major cell types for all cell types, (**c**) and mesenchymal

cells (**c**′). Size correlates to the percent of cells per cluster which express each gene. Color corresponds to the average expression of that gene within cells of that cluster (high expression=blue). **d**, **d**′ Module scores of hub genes from selected WGCNA modules calculated across major cell types via Seurat (ModuleScore). Color indicates average expression based on combinatorial expression from all genes of each module, across cells within each cluster (low = blue, high = red). Percent expression is indicated by size.

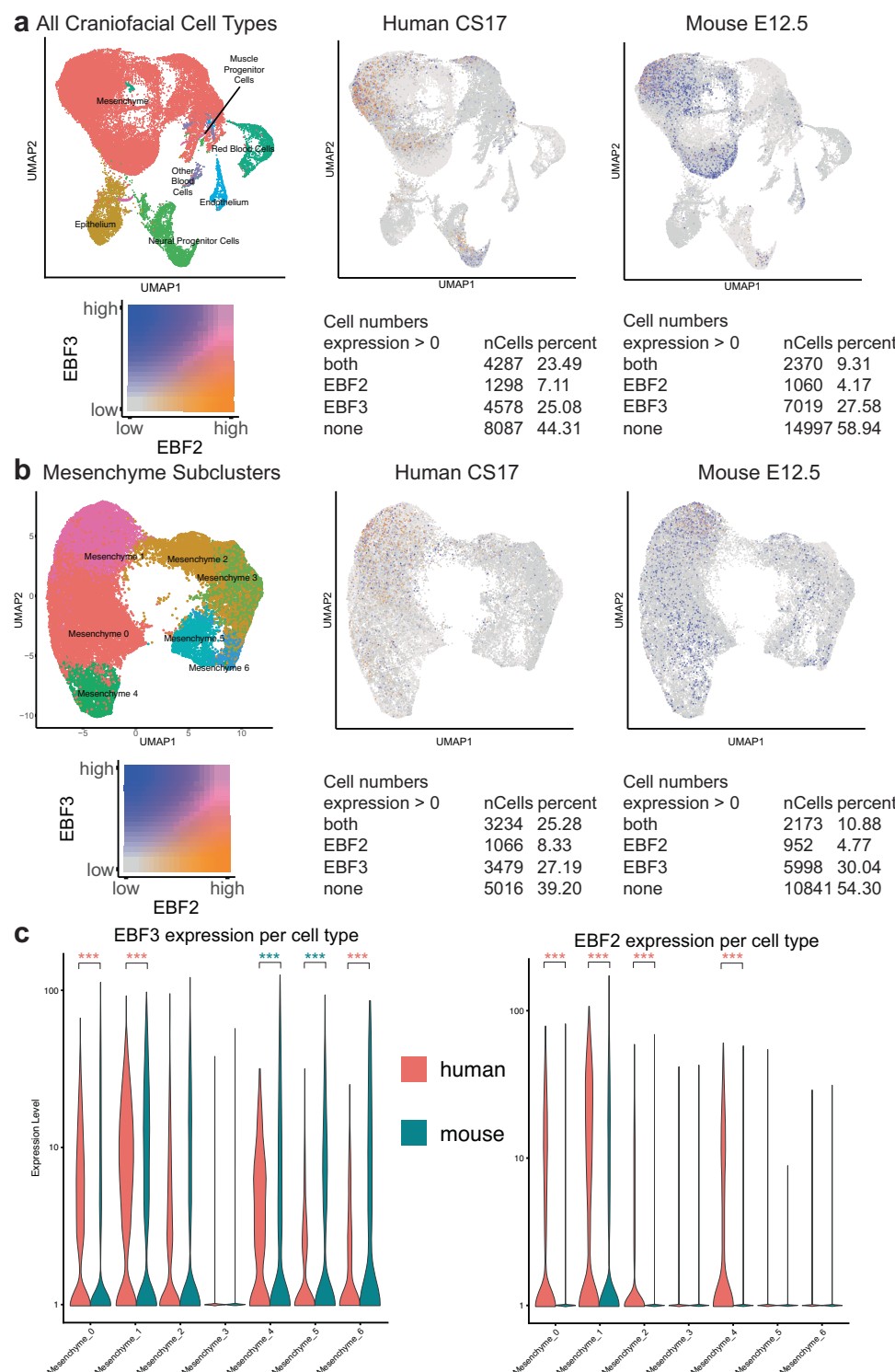

**Fig. 8 | EBF2/3 Show species-specific biases in mesenchymal cell subpopulations. a** Co-projected UMAPs of CS17 and E12.5 craniofacial tissues of all cell types. Expression plots of *EBF2* (orange) and *EBF3* (blue) in human and mouse demonstrate highly different rates of coexpression (pink) based on species (23.49 human, 9.31 mouse). **b** Co-projected UMAPs of mesenchymal subclustering of CS17 and E12.5 craniofacial tissues. Expression plots of *EBF2* (orange) and *EBF3* (blue) in human and mouse demonstrate highly different rates of coexpression (pink) based on species (25.28 human, 10.88 mouse). **c** Expression of *EBF3* in mesenchymal

subclusters shown in (**b**), split by species. Clusters 0, 1, and 6 demonstrate human-biased expression, while clusters 4 and 5 demonstrate mouse-biased expression. *EBF2* expression in mesenchymal subclusters displays human-biased expression in most clusters, with only cluster 6 showing bias towards mouse (***$p < 0.05$ for clusters with median expression >1 in at least one species based on Wald test performed by DESeq2[141], pink stars indicate human bias, blue stars indicate mouse bias).

dysostosis in Treacher Collins syndrome is linked to mutations in several ubiquitously expressed genes, including *TCOF1* and components of RNA Polymerase I, *POLR1C* and *POLR1D*. While such genes are clearly important for normal craniofacial development, they likely exert their effects at the post-transcriptional and/or translational levels, making them difficult to interrogate with the data that we have collected. However, transcriptomics is well suited to identify genes with highly specific expression. These CF Gini genes were enriched for genes known to be involved in CF disorders in humans and mice (Fig. 2b), and included many genes that were either not known to be expressed in CF development or not appreciated to have highly restricted expression. By integration with our previously described chromatin state segmentations[121], we demonstrated that these specifically expressed genes are systematically surrounded by craniofacial-specific enhancers over distances of ~400 kb. These genes represent excellent disease candidates and warrant further exploration by the field (Supplementary Data 7).

Analysis of timeseries differential gene expression revealed dynamic gene utilization that coincided with known morphological changes throughout development. Our work identified two subsets of dynamic genes that are enriched for craniofacial disease ontologies. Their trajectories of expression peak during 4 to 4.5 post-conception weeks (CS13-CS14) and again at 5.5 to 6 post-conception weeks (CS15-CS17) (Fig. 4a). The fusing of major facial prominences is believed to occur during this time period, the failure of which leads to CL/P. These pairwise analyses are crucial for helping to understand how gene expression changes over time; however, many genes that are important for craniofacial development with steady expression across our samples may be overlooked. A prominent example is *MSX2* which has robust expression in our data and is identified by the Gini index, but does not rise to significance across the time series. To address this issue, we employed coexpression analysis with a commonly used approach, WGCNA. WGCNA has been demonstrated to identify biologically relevant gene networks in a variety of contexts and allow for more nuanced complex gene expression dynamics to be detected.

The WGCNA network we constructed has several modules significantly enriched with genes involved in embryonic patterning, cell-signaling, and cell–cell adhesion (Fig. 5a). More specifically, we found two modules have enrichment for known craniofacial disease genes and one of these, the black module, is enriched for CF Gini genes and CFSE targeted genes. These findings prompted us to examine this module more closely for candidate disease genes. In particular, genes that are coexpressed with many other genes or potential "hubs" of regulation have been shown to be critical for model organisms[131,132].

Highly connected hub genes from coexpression analysis are also likely to be depleted of variants in otherwise healthy humans, and, thereby, excellent candidates for disease. We showed that genes with mutational constraint in the human population are enriched in the hub genes within our coexpression networks (Fig. 6a). However, it's important to note that not all identified hub genes are relevant within the system[133], so it is necessary to integrate other layers of data to identify candidates genes of craniofacial disease specifically. We addressed this by incorporating craniofacial-specific expression and putative craniofacial-specific enhancer targeting and examined all modules for these characteristics shared with the genes in the black module. In particular, we observed that the light green and dark slate blue modules contain many craniofacial-relevant hub genes, such as those in the *WNT* family and *PHOX2* genes. This allowed us to unbiasedly identify 539 genes, 239 of which are not known to be associated with any human craniofacial disease (Supplementary Data 7). We showed here this prioritized list was singularly enriched for de novo mutations in patients with orofacial clefting, particularly those genes observed in multiple patients. These findings strongly suggest others in this list of genes and the craniofacial regulatory regions that surround them are likely to be involved in orofacial clefting in humans.

Additional sequencing of expanded numbers of trios affected by orofacial clefting will be necessary to validate this result.

While we did not observe mutations of *EBF3* in the orofacial clefting trio data, this gene was our top prioritized gene. To explore this prediction, we examined the *EBF3* locus chromatin architecture, GWAS associations for orofacial clefting, phenotype data from haploinsufficient individuals, and cell-specific expression in both human and mouse. There are two GWAS variants associated with orofacial clefting in linkage disequilibrium with enhancers at the locus[122,123,134]. One is located in an intronic region of *MGMT* and therefore was predicted to affect that gene, however, *MGMT* is expressed across most tissues (Gini score 0.23, Fig. 3d). We contend here that *EBF3* is a better candidate due to its conspicuous signals from multiple genomic data types in our analysis. Indeed our analysis of additional patients with *EFB3* variants or copy number changes revealed a strong increase in orofacial clefting incidence. When we compared gene expression of both *EBF2* and *EBF3* between human and mouse craniofacial tissues from similar stages of development, we observed considerable differences in coexpression in two subtypes of mesenchyme (Fig. 8c). We observed the strongest expression of both genes in human mesenchyme subcluster 1. This subcluster is likely to be from the upper palate based on *RUNX1* expression (Fig. 7a) and could explain how *EBF3* is a risk factor for cleft palate. Given the strong coexpression of these two genes and similar chromatin state characteristics in human craniofacial tissue, *EBF2* may also be a risk factor for craniofacial disease that remains to be uncovered.

While we highlighted the expression of *EBF2* and *EBF3* in these data, these data represent the first survey of cell-type level gene expression during human craniofacial development. We find strong similarities in overall numbers and proportions of cell types between humans and mice at CS17 and E12.5, consistent with our findings in bulk gene expression (Supplementary Fig. 2E). It will be interesting to determine if there are species-specific differences in timing or type of cells generated during craniofacial development that could be linked to disease risk or the substantial phenotypic differences in skull shapes. Further exploration of gene expression patterns and chromatin accessibility of individual cell types across multiple stages of craniofacial development will be needed to better address this issue. It should also be noted that we have focused much of our analysis on mesenchyme. Neural crest cells are key contributors to the mesenchyme of the craniofacial region and defects in neural crest function are linked to a variety of craniofacial abnormalities. We show here at CS13 that the bulk expression pattern reflects that of cultured CNCCs (Supplementary Fig. 6D), suggesting that most of the mesenchyme at this stage is neural crest-derived. However, mesoderm is also a contributor to mesenchyme. Clearly defined mesoderm has been described in much earlier human embryonic stages (CS7)[135] and likely not present in substantial numbers at the stages we have investigated to influence bulk gene expression patterns. Further analysis of single-cell gene expression ranging from CS7 through at least CS17 would be needed to explore how this cell type contributes to human craniofacial development.

In closing, by integrating multiple layers of functional genomics data, including gene expression, chromatin states, chromatin structure, and population genetics, we have provided compelling evidence for additional disease genes in human craniofacial development. We provide all data in formats that are directly comparable to other large consortia and can be downloaded from Gene Expression Omnibus. We have updated our existing public track hubs on the UCSC Genome Browser and the Cotney Lab website. For ease of data exploration, we have provided the bulk and single-cell data in this study as interactive shiny applications http://cotneyweb.cam.uchc.edu/craniofacial_bulkrna/ and http://cotneyweb.cam.uchc.edu/craniofacial_cs17_e12_5/. We hope these resources can be used by the field to better understand human craniofacial development and aid clinicians in interpreting genetic data from craniofacial abnormality patients.

## Methods

### Human tissue samples

The use of human embryonic tissue was reviewed and approved by the Human Subjects Protection Program at UConn Health (UCHC 710-2-13-14-03). Human embryonic craniofacial tissues were collected via the Joint MRC/Wellcome Trust Human Developmental Biology Resource (HDBR) under-informed ethical consent with Research Tissue Bank ethical approval (18/LO/0822 and 18/NE/0290, project 200225). Donations of tissue to HDBR are made entirely voluntarily by women undergoing termination of pregnancy. Donors are asked to give explicit written consent for the fetal material to be collected, and only after they have been counseled about the termination of their pregnancy. Further documentation of all policies and ethical approvals for HDBR sample collection can be found at https://www.hdbr.org/ethical-approvals. Tissues were flash-frozen upon collection and stored at −80 °C. Upon thawing, the samples were quickly inspected for intactness of the general craniofacial prominences. They were further dissected to exclude other tissues that might have been included initially, such as eye spots or cardiac outflow tracts. The general regions that were collected are indicated in yellow in Fig. 1a. All CS22 samples were processed by the laboratory of Axel Visel and retrieved from Facebase (Supplementary Data 1)[34,41].

### Mouse embryonic tissue samples

The use of mouse embryonic tissues was reviewed and approved by the UConn Health Institutional Animal Care and Use Committee (Protocol AP-2000061-0723). Eight-week-old wild-type male and female C57BL6/J mice were obtained from Jackson Laboratory. Mice were housed according to recommendations by Jackson Laboratory with 12 h light:dark cycle beginning at 7 a.m. The ambient temperature was maintained between 20 and 22 °C and humidity was maintained at 40–60%. Mice were given ad libitum access to food and water. Timed matings were established by the identification of vaginal plugs the morning following the housing of a single male with multiple female mice. Embryos were harvested from pregnant mothers at mid-day 12 days after identification of the vaginal plug. The staging was confirmed by counting somites and comparing overall morphology to the Theiler Staging Criteria[136]. All embryos from a given litter were combined for individual biological replicates. Craniofacial prominences were collected in a very similar fashion to human samples and subsequently prepared for snRNA-Seq.

### CNCC differentiation

Dissociated (using 40um filter) H9 ESCs were plated at 30,000 cells per cm$^2$ in NCC media with 10 ul Rock Inhibitor and 10 uL of 3 mM stock CHIR99021. NCC Media: 48.5 mL DMEM/F12 (Gibco#10565-018) 500 uL 50 U/mL penicillin, 500 uL 50 U/mL streptomycin, 1 mL B27. Media was changed daily, the rock inhibitor was added on the day following plating and not after. Differentiation is complete by day 5.

### RNA-seq

Frozen tissue samples were added to Qiazol (Qiagen) and subjected to mechanical disruption using a motorized pestle. Homogenates were then processed using the miRNeasy RNA extraction kit (Qiagen, 217004) with on-column DNAse treatment (Qiagen, 79254) according to the manufacturer's protocol. RNA integrity was checked using Agilent Tapestation 2200 with Agilent RNA analysis screentapes (Agilent Genomics, 5067-5576). RNA with RNA integrity number (RIN) scores, preferably >8.0, were used in the preparation of RNA-seq libraries. RNA-seq libraries were prepared from 100–200 ng total RNA using the TruSeq stranded mRNA kit (Illumina, RS-122-2101) according to the manufacturer's instructions with the modification to use Superscript III Reverse Transcriptase enzyme (Invitrogen, 18080044) during the first strand cDNA synthesis step. Completed libraries were checked for quality and average fragment size using the Agilent Tapestation 2200 with D1000 screentapes (Agilent Genomics, (5067-5582). Molar concentration was determined using the NEBNext qPCR library quantification kit (NEB, E7630). Libraries were pooled and diluted to 1.8 pm and sequenced on the NextSeq500 Illumina platform using 75 bp paired-end sequencing according to the manufacturer's recommendations. Libraries were diluted to 4 nM, pooled, and denatured according to the instructions for Illumina NextSeq 550/500. Libraries were sequenced on the NextSeq 500 or 550 with settings for single-index, paired-end sequencing with 75 cycles per end.

### snRNA-seq

Samples were mechanically disrupted into liquid suspensions, checked for viability, and counted using Trypan blue staining. Nuclei were isolated and quantified following the established protocol (10x Genomics®). Samples were transferred to the Jackson Laboratories (Farmington, CT) Single Cell Biology Laboratory (SCBL) for processing which followed the *Chromium Next GEM Single Cell Multiome ATAC + Gene Expression* user guide from 10x Genomics®. Sequencing was done on an Illumina NovaSeq.

### RNA-seq data processing

**Human.** Quality control was performed on RNA-seq reads using FastQC (v.0.11.7) and MultiQC (v.1.1)[137]. Trimming for adapters, quality, and length was performed using Trimmomatic (v.0.36)[138]. Trimmed fastqs were aligned with Rail-RNA (v.0.2.4b)[45] using human assembly GRCh38/hg38. RSeQC (v.4.0.0)[139] was used to calculate the read distribution, gene body coverage and transcript integrity number (TIN) score[140] (Fig. 1b and Supplementary Fig. 1A). The coverage bigWig files output by Rail-RNA (v0.2.4b) were used as input for the generation of counts tables by following the instructions and pipeline from *recount2* (https://github.com/leekgroup/recount), where the comprehensive Gencode v.25 annotation was used. The *level 3* genes, as defined by gencode were excluded. The recount rse_gene objects for each sample were combined into one rse_gene object and transformed with scale_counts from recount (v.1.8.2). The PCA plots in Fig. 1b and Supplementary Figs. 1, 2 were made using the prcomp function from the built-in R (v3.5.3) stats package on a DESeq2 (v.1.26.0)[141] rlog transformation on the raw counts of the 18597 most highly expressed genes across all craniofacial samples generated in this study.

**Mouse.** The counts table for mouse data was collected and generated by the pipeline from Recount3[70] R package (v.1.8.0). 500 samples from 17 studies were gathered into a rse_gene object and scaled with transform_counts function of Recount3[70] (v.1.8.0). Gencode v.M23 annotation was used. The integrated count's table was further performed batch correction with RUVs in RUVSeq[142] R package (v.1.3.2) A full listing of samples used in this analysis is provided in Supplementary Data 1. After generating scaled and batch-corrected counts table, all pseudogenes and microRNA genes by Annotationdbi[143] (v.1.60.0) were discarded, thereby 26379 genes were used for downstream analysis. For the PCA plot in Supplementary Fig. 2, we used pcaplot3d function from pcaExplorer[144] package (v.2.24.0) and the plotly function from the plotly package (v. 4.10.1).

**Human and mouse craniofacial data comparison.** Trimmed fastqs of human craniofacial data (CS13–CS17) were aligned and annotated on GRCh38/Gencode v.26 with Monorail pipeline (v20220219) (https://github.com/langmead-lab/monorail-external) to build rse object in recount3[70]. Mouse facial data were from ENCODE (GSE37909, GSE57230), collected by the recount3[70] database (compatible SRA ID: SRP013027 and SRP013703). Each counts table was further performed with scaling and batch correction, respectively. To combine human and mice data, only 1:1 orthologs between hg38 and mm10 were used.

Thereby 15567 genes were used for every downstream analysis for human and mice craniofacial comparison, in log2-transformed values with offset 1.

## snRNA-seq data processing

Human raw fastqs were aligned to Gencode37 using CellRanger (v6.1.2) and gene counts per cell were imported into Seurat[145] (v3.2.0). Mouse raw fastqs were aligned to mm10 genome using CellRanger and gene counts per cell were imported into R. Each mouse mm10 counts table was converted to human Gencode37 orthologs using a 1:1 orthology table. Mouse and Human genes with no single ortholog were excluded from our analyses. Seurat[145] (v.3.2.0) was used for filtering, merging of samples, scaling, normalization, dimensionality reduction (UMAP), and clustering. Clusters were functionally annotated using GO enrichment analysis in clusterProfiler[146] (v.3.14.3) from marker genes of each cluster. Enrichment of the WGCNA module hub genes per cell type (Fig. 7b, c) were calculated using Seurat[145] function AddModuleScore (options bins = 1, control = 1000, seed = 1, search = TRUE). Coexpression UMAPs were calculated using FeaturePlot with blend =TRUE and order=TRUE. Detailed scripts of analyses can be found at our github (https://github.com/cotneylab/Embryonic_Gene_Expression_and_Chromatin_Dynamics/tree/master/Craniofacial_Transcriptomics).

## GTEx analysis

The rse_gene R data object, or counts table of all GTEx tissues was retrieved from the recount2 database (https://jhubiostatistics.shinyapps.io/recount/). The GTEx counts table was generated using the same Rail-RNA and recount2 pipeline that was used to generate the count tables for our embryonic craniofacial data which is described above. The GTEx dataset contained 9662 bulk RNA-seq samples which were combined with 24 embryonic heart samples[39], five fetal cortical plate samples[47], and 12 embryonic craniofacial samples. The metadata for GTEx is provided in a link under the phenotype column from the recount2 database, and the tissue assignments located under the column named *smts* was used, resulting in a total of 34 unique tissues. The counts were transformed using the scale_counts function from the recount (v.1.8.2) R package.

The plots in Fig. 1d were analyzed using a down-sampled version of GTEx, where twelve random samples from each tissue were chosen. The counts matrix was filtered for lowly expressed genes using rowMeans >50. A transformation was performed using vst function from DESeq2[141] (v.1.26.0). PCA plot was made for the top 5000 most variable genes using the prcomp function.

The tSNE plot in Supplementary Fig. 1D was made using GTEX, embryonic heart, fetal brain, and craniofacial samples counts table filtered for lowly expressed genes. The filtered counts matrix was transformed by log10 with a pseudo count of 1 added to all values. The transpose of the log10 transformed matrix was then converted to a distance matrix using the dist function in R. This distance matrix was used as input for the tSNE model generated by using the Rtsne function from the R package Rtsne (v.0.15). The parameters used in Rtsne were the following: perplexity = 100, max_iter = 1000, theta = 0.5, dims = 2.

For qualitative signal comparisons across tissues in Fig. 1c, the UCSC genome browser was used to load the GTEx RNA-Seq Signal track hub. Expression signals from 30 samples of each tissue type indicated were randomly selected. BigWigs generated by the Rail-RNA pipeline above for craniofacial samples were visualized alongside these tracks at the *PAX7* gene locus.

## Tissue-specificity of gene expression (GINI)

A total of 34 tissues, 31 from GTEx, and embryonic heart, craniofacial, and fetal brain was used to calculate the tissue-specificity. The Gini Index for each gene was calculated using the Gini function from R library Ineq (v0.2-13) on the average counts across all samples per tissue. The discrete function used by Ineq:

$$G = \frac{2\sum_{i=1}^{n} i \times x_i}{n \sum_{i=1}^{n} x_i} - \frac{n+1}{n}$$

Where $x$ is the raw average count of the gene for tissue, $i$ indexed such that $x\_i \le x\_(i+1)$, and $n$ is the number of tissues surveyed.

A gene was given a tissue assignment based on the tissue with the maximum average count across all tissues for that gene. To create the heatmap of most restricted genes, the distance matrix of one minus the transpose of the Pearson correlation of the average expression per tissue for all genes with a Gini score of 0.7 and above was clustered using hclust with method = "complete". For visualization purposes, in the heatmap of Fig. 2a, sex tissues, including breast, testis, vagina, cervix uteri, fallopian tube, prostate, ovary, and uterus, were excluded but are included in Supplementary Data 1. Testis are known to have a large amount of upregulated genes, thus likely contributing to the large amount of Gini genes assigned to it[36]. The average expression per tissue matrix was plotted using a pheatmap with scaled rows organized by the dendrogram calculated from hclust. Disease enrichment analysis was done using the R package disgenet2r[147] (v0.99.2) disease_enrichment function on genes 0.7 and above. The fold enrichment score was calculated by dividing the *Ratio* by the *BgRatio* columns and taking the log base 2.

## Craniofacial-specific enhancer effects on gene expression

Assignment of enhancers to genes was made using GREAT (v.4.0.4) on craniofacial-specific enhancers (CFSE) hg19 with the whole genome as background and using the single nearest gene association rule setting to 1MB. The line graph in Fig. 3a of the enhancers versus expression used the average expression per tissue type from GTEx and our heart enhancer data using geom_smooth from ggplot2 (v3.3.1) with the method set to "gam".

## Distance to craniofacial-specific enhancers analysis

For Fig. 3b, we used Gencode annotation v25 to get the location of TSS for all genes. A Gini gene was defined as having ≥0.7 Gini score. Bedtools[148] (v.2.28.0) closest was used to get the distance between the closest Gini TSS to CFSE. A background distribution was established by picking the distance to a random set of 239 Gini genes among all tissues (n = 33), excluding those assigned to CF at 1000 permutations. The interpolation lines were plotted using geom_smooth with formula=y-log(x). To get the median closest gene-enhancer for our data, we only considered enhancer pairs within 400 kb distance as that was significant over random.

Supplementary Fig. 3A, B use the ref. 149 data. We plotted a histogram for their 664 high-confidence enhancer gene pairs to compare directly to Fig. 3b. Supplementary Fig. 3B is a violin plot of these same 664 high-confidence enhancer gene pairs and our CF Gini- CFSE pairs that were ≤1 Mb (n = 3437).

## Differential expression

The scaled rse_gene recount object including NCC, CS13, CS14, CS15, CS17, and CS22 samples were made into a DESeq2[141] (v.1.22.2) object. Low gene counts were filtered by removing all genes whose mean of counts across all samples was less than 50. This left a total of 18,597 genes for downstream analysis. Batch effects were mitigated by using the sva (v.3.30.1) package in R. Included the CS13, CS14, CS15, CS17, and CS22 for the sva calculation, which resulted in four surrogate variables. Pairwise differential analysis between CS13, CS14, CS15, and CS17 were performed with DESeq2[141], including the surrogate variables. Differentially expressed genes were plotted in the heatmap of Fig. 4a considered if they had a Benjamini−Hochberg-adjusted p value of less than

0.05 and a log2 fold change of greater than absolute value of 1. The PCA plot in Supplementary Fig. 6A was made as described above, except on just the 3677 differentially expressed genes. The R package Venerable (v.3.1.0.9) was used to make Supplementary Fig. 6B.

The heatmap in Fig. 4a was made using the R package pheatmap (v.1.0.12) on the rlog transformation of the raw counts. The rlog counts distance matrix of one minus the transpose of the Kendall correlation was clustered using hclust with method = "complete". The counts were scaled by setting the pheatmap scale option to "row". The resulting hierarchical clustering was used in the pheatmap option cluster_rows to organize the gene rows and the hierarchical clustering of the sample columns was from pheatmap default options.

ClusterProfiler[146] (v3.12.0) was used to obtain the gene and disease ontology enrichments from the dendrogram clusters (Fig. 4b and Supplementary Fig. 4E) using the function enrichGO with standard options. The fold enrichment score was calculated by dividing the *Ratio* by the *BgRatio* columns and taking the log base 2.

In Fig. 4c, the disease ontologies were obtained for the light green (cluster 2) differentially expressed genes using enrichDGN. The network plot was made using cnetplot from enrichplot (v.1.6.1) using log2 fold change between CS13 and CS17 from DESeq2[141].

The heatmap in Supplementary Fig. 6D was made as described above, utilizing six samples from GEO accession GSE70751, SRX numbers SRR2096446 through SRR2096451 linked to (ref. 42). The fastqs were processed using the same Rail-RNA/recount pipeline as all other data in this paper.

## Gene class analysis of differentially expressed genes

Three lists were used to represent most transcription factors[57] ($n = 1639$), RNA-binding proteins ($n = 415$), and genes annotating the GO biological process cell–cell signaling, GO:0007267 ($n = 1364$). Trajectory plots in Supplementary Fig. 7A, B were made using geom_smooth from ggplot2 (v3.3.1). The quantitative locations of the x-axis depicting timepoints were determined by taking the median x-value for each timepoint from the PCA plot of Fig. 1b. Enrichment over background (Supplementary Fig. 7C) was determined by randomly selecting 3677 genes from the 18,597 used in the DESeq2[141] analysis 1000 times.

## ChIP-seq data integration

**Across differentially expressed genes.** For Supplementary Fig. 8: Imputed chromatin signals for each histone mark and DNAse hypersensitivity were extracted in a 5 kb window surrounding the transcription start sites of genes from each indicated cluster using computeMatrix reference-point command in deepTools[150] (v.3.5.1). Histograms and heatmaps were generated using the output of computeMatrix in plotHeatmap command of deepTools.

**Differential H3k27ac signal.** For Supplementary Fig. 9A: H3K27ac counts at each craniofacial enhancer region were generated from bigWig files of imputed H3K27ac −log10($p$ value) signals using rtracklayer[151] (v.3.17) and GenomicRanges[152] in R (v3.5.3) based on an original read length of 75 bp. Heatmap constructed from H3K27ac counts filtered for those determined to be significantly different between CS13 and CS17 using DESeq2 for the following cutoffs: Benjamini–Hochberg $p$ adjusted <0.01 and a log2 fold change > abs(1). The counts distance matrix of one minus the transpose of the Pearson correlation was clustered using hclust with method = "complete". The dendrogram was cut into two clusters and the H3K27ac segments from each cluster were intersected with genome-wide TSS using bedtools[148] (v.2.28.0) The permutation test to establish a background of intersecting genes was done by randomly selecting segments from the genome using bedtools[148] shuffle with the -chrom setting. The genes of these TSS's were intersected with the differentially expressed genes from Fig. 4.

## Weighted gene coexpression network analysis

We generated coexpression networks using the WGCNA[106] R package (v1.69-80) based on recommendations put forth by the Horvath group (https://horvath.genetics.ucla.edu/html/CoexpressionNetwork/Rpackages/WGCNA/).

**Network construction.** The rse_gene object from the recount pipeline was combined for all samples resulting in a starting matrix of all genes (Gencode v.25) by 17 samples, including three replicates of CNCC, CS13, CS14, CS15, and CS17 and two replicates of CS22, the raw counts were scaled by using scale_counts from recount (v1.8.2). Lowly expressed genes were filtered by excluding all genes whose sum across all samples were lower than 100. SVA was used to transform the counts to account for batch effects, four surrogate variables were detected. All resulting negative counts were set to 0. The counts' matrix was further transformed by log2 with a pseudo count of 1.

A soft-thresholding power of 18 was chosen assuming a signed network and based on recommendations for less than 20 samples. The modules were detected from the network from the cutreeDynamic function from the WGCNA package with the following parameters, minClusterSize = 100, deepSplit = 2. Detected modules were merged based on their eigengene correlation. To do this, a dendrogram of the module eigengenes was generated and a threshold value of 0.18 was chosen as input for the function mergeCloseModules. The resulting network contained 29 modules. The intra-modular connectivity of each gene was calculated using the intramodularConnectivity function from the WGCNA library, which was the metric used to determine the hub and non-hub designation.

**Plotting of modules.** A multidimensional scaling of the module eigenvectors output from WGCNA[106] was generated to plot the modules in 2D space using the function cmdscale from the stats (v.3.6.1) R package. A Pearson correlation of the module eigenvectors was calculated for the edges. Positive correlations of 0.5 and greater were included. Modules were plotted that fulfilled the criteria of having significant adjusted $p$ values (<0.05) from the GO analysis, or significant (Benjamini–Hochberg) adjusted permutation $p$ values (<0.05) in at least two of the gene lists from the Tissue-Specific, gnomAD, or DisGeNET categories (Fig. 5c). The module eigenvectors of each of the 17 modules were plotted using the geom_smooth function with method = "loess". The quantitative locations of the x-axis depicting timepoints were determined by taking the median x-value for each timepoint from the PCA plot of Fig. 1B. The confidence intervals were removed for ease of visualization.

Fig. 6c was made using the open-source software Cytoscape[153] (v.3.8.2).

**Gene ontology and functional enrichments.** RDAVIDWebService[154] (v1.22.0) was used to obtain gene ontology enrichment of the genes within each of the 29 WGCNA[106] modules. The gene background list used was all the genes input into the WGCNA[106]. The module enrichments of gene lists were determined by a permutation test. The 26,626 genes were randomly scrambled among the same number and size of modules as the original network and overlapped with the respective gene lists. This was repeated for 1000 iterations to calculate a permutation $p$-value adjusted by Benjamini–Hochberg approach.

**Protein–protein interaction analysis.** To generate the ppi histogram (Fig. 6a), 100 randomized versions of the WGCNA network were made. This was done by randomly assigning the 26,626 genes to 29 modules of equal gene sizes to the original network using the R function sample. The ppi enrichment of up to 500 randomly chosen genes for each module of each of the 100 randomized versions was then determined using the STRINGdb[155] (v1.24.0) package. Up to 500 genes were used due to constraints from the STRINGdb[155]

package. The ppi database was loaded by using STRINGdb$new with version = "10" and score_threshold = 0.4. For each iteration, the output *p* value of the STRINGdb call get_ppi_enrichment was adjusted using the Bonferroni method. The number of modules that met the adjusted *p* value cut off of 0.05 was counted for each iteration to produce frequency values.

**Hub gene LOEUF enrichment.** For the LOEUF enrichment analysis among the hub vs non-hub genes (Fig. 6b), the non-hub genes were randomly sampled using the R function sample. The number of non-hub genes sampled were the same number of total hub genes within the network, the 10% with the highest connectivity of 26,626 genes or 2663 from the non-hub gene list. This process was iterated 1000 times to get a mean and standard deviation for each LOEUF decile. The LOEUF score and decile designation for each gene is freely available through gnomAD v2.1.1

**Prioritized gene list.** The following parameters were used to generate Supplementary Data 7. To be considered for prioritization, a gene must have a LOEUF decile score of ≤3. It must then also meet at least two additional criteria: CF Gini gene, CFSE targeted, or WGCNA hub gene. Overlaps of genes with de novo mutations reported by Bishop et al.[126] were calculated with GeneOverlaps in R (version 1.34.0).

**Analysis of whole genomes from trios with orofacial clefts**
This work was approved by the institutional review board of Emory University (protocol IRB00098814). Whole genome sequencing was performed at the Center for Inherited Disease Research at Johns Hopkins University (Baltimore, MD) and the Broad Institute as described previously in refs. 126,127. Sequence and phenotype data were available from the Database of Genotypes and Phenotypes (dbGaP) under study accession phs002220.v1.p1, phs001168.v2.p2, phs001420.v1.p1, and phs000094.v1.p1. De novo variants were identified as described previously in ref. 126. Annotation of high-confidence DNMs was completed with ANNOVAR (version 201707). Overlaps of genes with de novo mutations reported by Bishop et al.[126] were calculated with GeneOverlaps in R (version 1.34.0).

**Statistical analysis of patients with *EBF3* sequence and copy number variants**
**Human subjects research.** This work was approved by the institutional review board of Baylor College of Medicine (BCM protocol H-47546) and was conducted in accordance with the ethical standards of this institution's committee on human research and international standards.

The Baylor Genetics clinical database is a private database that contains the results of exome sequencing studies performed on a clinical basis. Coded information from this database was accessed and reported here in accordance with BCM protocol H-47546. Informed consent was not obtained from individuals or families since the research team had access only to data from which all identifying information was replaced with a number.

The DECIPHER database is an online publicly accessible database (https://www.deciphergenomics.org/) used by the clinical community to share and compare phenotypic and genotypic data. The DECIPHER database contains data from patients who have given consent for broad data-sharing. No additional consent was obtained since only aggregate data is being reported in this manuscript.

**Statistics.** For comparisons of the incidence of orofacial cleft between populations, we performed two-tailed Fisher's exact tests using a 2 × 2 contingency table calculator available through GraphPad QuickCalcs (https://www.graphpad.com/quickcalcs/contingency1/). To calculate the 95% confidence intervals of orofacial clefting in various populations, we used the binomial "exact" calculator available through the UCSF Clinical and Translational Science Institute (http://www.sample-size.net/confidence-interval-proportion/).

**Reporting summary**
Further information on research design is available in the Nature Portfolio Reporting Summary linked to this article.

## Data availability
The craniofacial bulk and single-cell RNA-seq data generated in this study have been deposited in the Gene Expression Omnibus (GEO) database under accession code GSE197513 and Database of Genotypes and Phenotypes (dbGAP) under accession code phs002008. The raw data were available under restricted access for human genomics data privacy concerns, access can be obtained by application to dbGap. Sequence and phenotype data for orofacial clefting whole genome sequencing is available from the (dbGaP) under study accessions phs002220.v1.p1, phs001168.v2.p2, phs001420.v1.p1, and phs000094.v1.p1

The single nuclei and bulk RNA-Seq data can be interactively explored at http://cotneyweb.cam.uchc.edu/craniofacial_cs17_e12_5/ and http://cotneyweb.cam.uchc.edu/craniofacial_bulkrna/ respectively.

Mouse craniofacial gene expression generated by ENCODE were retrieved from GEO through accessions GSE37909 and GSE57230. A full listing of all data utilized from all 500 mouse samples retrieved from Recount3 is available in Supplementary Data 1.

Human CNCC RNA-Seq data were obtained from GEO accession GSE70751, SRX numbers SRR2096446 through SRR2096451.

## Code availability
All original code has been deposited in the Cotney Lab GitHub (https://github.com/cotneylab/Embryonic_Gene_Expression_and_Chromatin_Dynamics/tree/master/Craniofacial_Transcriptomics) and is publicly available.

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

## Acknowledgements

We would like to thank members of the UConn/JAXGM Single Cell Genomics Core for help with standardizing single-cell isolation techniques and preparing sequencing libraries. We would also like to thank members of the UConn Computational Biology Core and High-Performance Computing Facility for assistance with package installation and software/hardware support. This work was funded by grants from the National Institutes of Health to JC (NIDCR 1R01DE028945, NIDCR 1R03DE028588, and NIGMS 5R35GM119465).

This study makes use of data generated by the DECIPHER community. A full list of centers who contributed to the generation of the data were available from https://deciphergenomics.org/about/stats and via email from contact@deciphergenomics.org. We note that those who carried out the original analysis and collection of the DECIPHER data bear no responsibility for the further analysis or interpretation of the data.

## Author contributions

Conceptualization: T.N.Y. and J.C. Investigation: A.W., T.G., J.A.R., E.W.W., and J.V. Formal analysis: T.N.Y., S.O., T.G., D.A.S., E.W.W., K.R., E.J.L., and J.C. Writing—original draft: T.N.Y. and J.C. Writing— review and editing, T.N.Y., E.W.W, S.O., A.W., T.G., J.A.R., D.A.S., K.R., J.V., E.L., and J.C. Funding acquisition: J.C. Supervision: D.A.S., E.L., and J.C.

## Competing interests

The authors declare no competing interests.
