## [Peer Review File · Nature Communications]

Integrative analysis of transcriptome dynamics during human craniofacial development identifies candidate disease genesREVIEWER COMMENTS

Reviewer #1 (Remarks to the Author):

This is an interesting manuscript that generated and analyzed bulk and single cell RNA-seq on human craniofacial tissues from embryonic development 4 to 8 post conception. They integrated RNA-seq data from replicates of early-stage hESCs and H9 hESC-differentiated cranial neural crest cells (CNCCs), primary craniofacial tissue (CS13, 14, 15, 17), and late-stage CS22 data (Pennacchio et al., 2017). They were able to demonstrate significant overlap of upregulated genes from the early-stage cultured cells and primary CS13 tissues, and they also used PCA to demonstrate that craniofacial tissues generally clustered together and distinct from other GTEx tissues. They perform several analyses showing putative: 1) gene expression profile of 239 genes specific to craniofacial tissues, 2) ~400 kb location of enrichment of enhancers as potential regulatory chromatin interaction sites and DNA-binding factors with bivalent promoter status during CF development with strongest trends in both numbers of craniofacial specific strong enhancer segments and specificity of expression, and 3) coexpression networks with hub genes resistant to mutation in normal healthy populations.

There are several major strengths of this manuscript:

1. This is a study of early human craniofacial tissues, which are difficult to obtain for research study and of clinical relevance.
2. They perform several quality control metrics to validate their datasets.
3. They perform multiscale analyses integrating their RNA-seq data with human and mouse GWAS, WGS, dbGAP, ChIP seq and Roadmap Epigenome datasets.
4. Their analyses reveal many novel disease candidate genes for craniofacial development and associated abnormalities that can be studied in the future by the scientific community.

There are several major weaknesses of this manuscript:

1. They do not describe in enough detail the samples and tissues obtained and which specific craniofacial structures and tissues were dissected for use in these studies.
2. They do not describe the datasets of known craniofacial genes they use to determine enrichment of craniofacial specific genes. Are there truly many genes that are craniofacial specific? Most genes and pathways are expressed in multiple tissues to a varying degree at different times in development. Perhaps these genes should be considered enriched and not specific to craniofacial tissues. What criteria were used for cut off to select the 239 craniofacial gene with the highest average of expression level from the 5000 Gini genes? Why not 250 craniofacial genes with the highest average of expression?
3. They validate that their analyses by finding enrichment of genes and regulators known from the literature to be relevant to human and mouse craniofacial development, but they do not perform any functional studies. Although their data analyses suggest novel human craniofacial candidate genes involved in embryonic development and disorders, they do not prove that any of these novel genes indeed are involved. This would require at a minimum either in vitro cell culture, ex vivo organoid, in vivo animal functional experimentation or detection of pathogenic variants associated with craniofacial conditions for which no disease gene has been previously assigned. There are WGS datasets of patients with craniofacial abnormalities including cleft palate available through the NICHD Gabriela Miller Kids First Pediatric Research Program that could be analyzed for variants in their candidate genes. Thus, their findings are generally descriptive and qualitative.

Examples of other issues:

1. Title does not reflect the integration of chromatin analyses for enhancers, transcription factors, etc. that were performed.
2. Page 1, references cited, but not in reference list: NIDCR, 2018; OMIM, 2021.

3. Page 2, para 1, line 8: delete period after variation.
4. Figure S1A is not labeled as such.
5. Figure S1B shows that one of the CS13 replicates is an outlier from the other CS13 samples, while all other samples cluster with their respective replicates. Why do the CS13 samples vary in gene expression profiles more than the other samples and is that of concern for further analyses?
6. Page 3, line 4 from the bottom, Figure S1D cited is not correct and should be Figure 1D.
7. Page 4, para 2, what databases of CF abnormalities and cleft palate and analyses were performed to determine that ...239 genes demonstrated the highest average expression in our craniofacial samples ...were most enriched for CF abnormalities and cleft palate...also should be rewritten by adding words ...they were most enriched for genes implicated in CF abnormalities and cleft palate.
8. Page 4, heading Genes expressed...abnormalities, para 3, lines 4 and 5, superscript font should be used instead of caret and Gou et al., 2015 should not be italics.
9. Page 4, heading Genes expressed...abnormalities, para 4, line 3, midline facial hypoplasia is not a correct phrase to describe facial features of Stickler syndrome. They should use midfacial hypoplasia. There is an extra parenthesis after Kondo et al., 2016.
10. Page 4, heading Genes expressed...abnormalities, para 4, lines 5-8, they incorrectly state that Russell Silver was underappreciated in CF development. If they look at the Clinical Synopsis section of the OMIM entry there are several facial features (macrocephaly, delayed closure of anterior fontanel, large anterior fontanel, triangular face, prominent forehead, micrognathia, retrognathia, downturned corners of mouth, thin lips) noted for this condition that are well known to clinicians.
11. Supplementary tables are not labeled as Table S1, S2, etc. so it is difficult to figure out the relevant datasets being cited throughout the manuscript. Is Table S1 available?
12. Page 6, para 2, lines 3-4, expression profiles of their differentiated CNCC samples do not cluster with other published CNCC from the literature (Prescott et al. , 2015) why is that?
13. Page 6, para 3, lines 2-3, Figure 4A should be cited.
14. Page 7, full para 3, line 3 and elsewhere in the text, superscript font should be used instead of caret.
15. Single-nuclei RNA-sequencing section was very descriptive about only major cell populations. They do not perform spatial-temporal analyses.
16. References list: articles do not have consistent format and other concerns such as Consortium, G. and GTEx Consortium (2017) has incorrect authors listed; Hansen et al. has year in incorrect order; Hsieh et al. does not have volume or page numbers; Lopes et al. and Reynolds et al. do not have page numbers; Taigman et al. may have incorrect authors' names.

Reviewer #3 (Remarks to the Author):

Yankee et al 2022, carry out an original investigation examining gene expression in human samples across key timepoints in craniofacial development. Bulk RNAseq is performed on tissue from different embryonic stages as well as from cranial neural crest cells derived from human embryonic cells, and this data is analysed for specificity of expression, overlap with disease and intersection with craniofacial specific regulatory elements. They demonstrate through differential gene expression analysis that there are dynamic expression changes across time, and clusters of genes are linked to GO terms, disease enrichment and chromatin states. The authors also used a gene co-expression analysis to identify modules of genes from the bulk RNAseq dataset. These were then characterised by GO analysis, expression trajectories and intersection with relevant gene lists to identify enrichments related to morphogenesis, disease and CF-specific expression. Candidate disease genes, including MAB21L1 and EBF3 were highlighted, supported by phenotypic observations in recently identified human disorders. The latter gene was further investigated in terms of

gene regulation and linked to two GWAS clefting hits. Finally, single nuclei RNAseq analysis of craniofacial tissue is used to assess cell-specific expression of EBF3 and the correlating network gene EBF2.

This is a comprehensive analysis of difficult to study human embryonic material, that sheds significant light on gene expression during key stages of craniofacial development. The authors have considerable expertise in the bioinformatic analysis and integration of datasets to extract meaningful information and predictions of genes and loci that are candidates for craniofacial disorders. Altogether, this work adds an important resource for the field, and provides a basis for ongoing investigations into craniofacial disorders relevant to both developmental biologists and clinicians.

I have a few considerations, and also minor comments below:

1) Tissue description. I think it is important that the tissue sampled is fully described. Does it correspond to the yellow highlighted regions in Fig1A? This isn't made clear. In general, this is important for the quality of the comparisons that are made between tissues and datasets. For example, was the tissue isolated for snRNAseq at CS20 equivalent to material isolated for bulk RNAseq? Similarly, does this apply to tissues used previously for epigenomic analyses?

I think there are also issues comparing between human and mouse: snRNAseq, CS20 versus E11.5. What is the evidence that these are the same stage for comparison? Also, the tissue isolated for the mouse E11.5 data is from a specific CF region, the lambdoidal junction. This is a small part of the overall craniofacial region. Is it viable to make comparison to this data when the tissues are likely not equivalent in time and space?

2) The snRNAseq data analysis feels under exploited and like a skim of the information that must be within this dataset, and its a bit strange that it is used simply to investigate EBF2/3 expression. I think that could be better accomplished through in situ analysis of comparable human and mouse craniofacial tissue.

I don't find the network enrichment analysis in relation to the snRNAseq data convincing, without in situ validation of selected genes. The mesenchyme cluster (the vast majority of the cells) is likely composed of multiple individual subtypes so stating that there is enrichment for specific cell types would perhaps be better supported by subclustering of the mesenchymal cell population. TWIST1/PRRX2 are stated to have similar expression in mesenchyme between species, however, from the data presented this is really hard to judge. If the mesenchyme was subclustered, and expression correlated with the top marker genes for each subcluster, this might help to assess these correlations. The same is true for the module score analysis, where there do appear to be zones of expression, particularly in the mouse mesenchyme.

In the human snRNAseq data EBF2/3 appear to be expressed in a subset of mesenchymal cells. What are these? There is some expression in the mouse cells – are these the same cells? Again, subclustering might clarify this.

It doesn't look like there is much expression of Ebf2/3 in the E11.5 data, but I'm not sure its informative to compare to the E13.5 Allen data to confirm expression patterns, which is 2 days later in development (even then I think there is mesenchymal expression at E13.5). It would be better to have in situ analysis of Ebf2/3 at E11.5 (or equivalent stage to CS20). Overall, the data only suggest differences in expression, which might actually be due to differences in the tissue sampled, and the stage of development.

This was a multiome analysis. It would be interesting to look at correlations between open chromatin and specific gene expression. Do the mesenchymal (subsets) ATACseq peaks correlate with the other epigenetic data in candidate genes and their expression? How does the data relate to the bulk CS22 data etc

3) Care needs to be taken in discussing mouse-human differences in EBF3 mutation outcome and concluding that that's there are different functions or expression patterns. The mouse mutants do not have overt craniofacial defects, but the phenotype in humans is not that strong in terms of craniofacial findings. Subtle craniofacial effects can easily be missed in a mouse mutant. Conversely, there are clear overlaps in the mouse and human phenotype (eg Blackburn 2017).

4) Mesoderm also contributes to craniofacial development, was this seen in the data? CNCCs are rightly used in the study, with a significant overlap with CS13, but I wondered if there should be a comment relating to mesodermal involvement.

Minor points

Line 93/94 – As pcw is used in the summary, but Carnegie stages used subsequently, it would make things clearer if the equivalent pcw range is included here at the start of the results.

Line 95 – these same stages – not CS22?

Line 98 – should reference to Figure 1B be here?

Line 100 – why is one of the CS13 samples so different in the staging PCA plot?

Line 133 – Figure 1D rather than S1D?

Line 151 – what are the 239 genes, and what genes are within each disease category? List in table S2?

Line 158 –the transcription factor enrichment – can these genes also be listed in table S2? Similarly, what are the genes within each category?

Line 164 – S2A should be S2. The Figure S2 figure legend – disease(?) categories

Line 196 – source of embryonic heart data? What stages were used?

Line 202 – I was confused as to why Fig 1D is referred to here. Is the enrichment data shown anywhere?

Line 206 – insert reference to Figure 3B. Fig S3A,B?

Line 234 – CFSE track for MSX2 is not shown

Line 266 – progressive dampening over developmental time – expression does seem to come back on in CS22

Line 273 – move S6E reference to line 276

Line 281 – The PAX3 and SOX9 trajectories don't look that different from the overall trajectory

Line 299 - , suggesting

Line 323 – 29 modules, but only 17 shown in Fig 5A?

Line 397 – in Fig 6C the diamond shape in the inset box is unclear

Line 405 – references for these previous associations?

Line 413 – paralog?

Line 415 – Why was CS20 chosen, and not one of the timepoints used in the RNAseq analysis?

Line 415 - How many nuclei were isolated/sequenced from the 2 samples?

Line 416 - What were the marker genes used for clustering into the 6 main cell types?

Line 427 – FigS9C -> FigS11C?

Line 429 – Please give more information on the module score – how its calculated (methods) and what it shows. Enrichment does seem to be in the mesenchymal population, but how does that look when these populations are subclustered to reveal the different constituent populations?

Line 433 – MAB21L1/Mab2111 expression cannot be described as cell-type restricted – it appears to be expressed in multiple cell types in both human and mouse.

Line 434 - FigS9D -> FigS11D?

Line 435 – Are these really comparable populations of cells (both spatially and temporally) to state that there are striking differences?

Line 439 – What human stage of Cf development does E13.5 correspond to? The scRNAseq data is from E11.5, perhaps expression patterns have changed?

Line 441 – TWIST1 and PRRX2 in Fig 7D look to me to be expressed in different cell types within the mesenchymal cluster, with some overlap in the PRRX2 domain.

Line 444 – lowercase for mouse genes in Fig S11E

Line 444/445 – There does appear to be Ebf3 in craniofacial mesenchyme. What about Ebf2? The statement that the data suggests the 2 genes have been removed or gained in humans is too strong

Line 480 – Fig4C?

Line 481 – changes over time?

Line 496 – genes with mutational constraint?

Line 511 – MGMT previously stated to be broadly expressed in the data? This also appears to be the case in Fig3D...

Line 770 – Include a description of the dissection describing exactly what tissue was analysed

Line 798 – Is this exactly the same tissue as analysed in the bulk RNAseq? There is minimal information on the protocol used. Were the 2 samples treated at the same time? Were there technical replicates? When were the samples combined? Presumably they were separate – do they overlap in UMAP plots?

Line 856 – Figure S1F?

Line 883 – Table S2?

Line 900 – For Figure 3B, we used..

Line 916 – Included the....?

Table S1 – Where is CS22 for bulk RNAseq?

REVIEWER COMMENTS

We would like to thank the reviewers for their time considering our manuscript and their thoughtful suggestions on how to improve our work. In particular we have now performed extensive new analysis of mouse development and identify genes with conserved craniofacial specificity. This new analysis has allowed us to identify the most closely related stages of mouse and human craniofacial development. Based on these findings we have now compared CS17 human and mouse E12.5 single nucleus RNA seq in new figures 7 and 8. We have now also collaborated with two other groups to add disease relevance to our finds. Specifically, we now show that EBF3 is indeed an orofacial clefting risk gene. We now also demonstrate that our multilayer gene prioritization scheme is strongly enriched for genes with multiple de novo mutations in orofacial clefting trios. We feel these substantial additions have improved the manuscript and addressed the reviewers' concerns. We address each specific comment below. We also provide a version of the manuscript that highlights all version in this revision compared to the original submission.

Reviewer #1 (Remarks to the Author):

This is an interesting manuscript that generated and analyzed bulk and single cell RNA-seq on human craniofacial tissues from embryonic development 4 to 8 post conception. They integrated RNA-seq data from replicates of early-stage hESCs and H9 hESC-differentiated cranial neural crest cells (CNCCs), primary craniofacial tissue (CS13, 14, 15, 17), and late-stage CS22 data (Pennacchio et al., 2017). They were able to demonstrate significant overlap of upregulated genes from the early-stage cultured cells and primary CS13 tissues, and they also used PCA to demonstrate that craniofacial tissues generally clustered together and distinct from other GTEx tissues. They perform several analyses showing putative: 1) gene expression profile of 239 genes specific to craniofacial tissues, 2) ~400 kb location of enrichment of enhancers as potential regulatory chromatin interaction sites and DNA-binding factors with bivalent promoter status during CF development with strongest trends in both numbers of craniofacial specific strong enhancer segments and specificity of expression, and 3) coexpression networks with hub genes resistant to mutation in normal healthy populations.

There are several major strengths of this manuscript:

1. This is a study of early human craniofacial tissues, which are difficult to obtain for research study and of clinical relevance.
2. They perform several quality control metrics to validate their datasets.
3. They perform multiscale analyses integrating their RNA-seq data with human and mouse GWAS, WGS, dbGAP, ChIP seq and Roadmap Epigenome datasets.
4. Their analyses reveal many novel disease candidate genes for craniofacial development and associated abnormalities that can be studied in the future by the scientific community.

We appreciate the reviewer recognizing all of these novel findings in our data and the value of the datasets to the craniofacial community. We have now added additional analysis of mouse data, more comparable timepoints of single nucleus RNA-seq, and integrated new human genetic data relevant for orofacial clefting.

There are several major weaknesses of this manuscript:

1. They do not describe in enough detail the samples and tissues obtained and which specific craniofacial structures and tissues were dissected for use in these studies.
 - The tissues were sourced from HDBR, a tissue bank located in the UK established in 1999 which ethically collects gestational material between 3 and 20 weeks of development. In our methods we

provide the link to their official website (www.hdbr.org). They have developmental biologists on site which stage, perform dissections, and flash freeze the material for the end user. We added further details in our methods describing steps we take upon thawing of the material including removal of extraneous tissue.

2. They do not describe the datasets of known craniofacial genes they use to determine enrichment of craniofacial specific genes. Are there truly many genes that are craniofacial specific? Most genes and pathways are expressed in multiple tissues to a varying degree at different times in development. Perhaps these genes should be considered enriched and not specific to craniofacial tissues. What criteria were used for cut off to select the 239 craniofacial gene with the highest average of expression level from the 5000 Gini genes? Why not 250 craniofacial genes with the highest average of expression?

- We agree that the terminology “craniofacial specific” may be misinterpreted. We used this term in the manuscript to indicate that we applied a formal measurement of statistical dispersion (the Gini Index) on the expression of genes across many tissues. However, a high Gini score does not also mean that the gene is only expressed in one tissue, which would indicate true specificity. We changed the language to “biased” throughout the manuscript to reflect this.
- The criteria used for the “Gini” gene cut off was based on dividing the genes’ gini scores (observations) into quartiles. Genes whose scores were in the first quartile were labeled housekeeping, and those in the top quartile are labeled as “gini” or overrepresented in a small number of the surveyed tissues. These details have been added to the main text and in the methods. Additionally, the density plot showing the distribution of gini scores has been added to supplemental figure 2.
- We have addressed the question of the top 250 genes in craniofacial tissue based on absolute expression in the tissue. We find these are strongly enriched for housekeeping genes and provide little specificity for craniofacial related biology. We report these findings in Results, section “Genes expressed specifically during craniofacial development are enriched for craniofacial abnormalities”, paragraph 2, “In contrast to the Gini index analysis, using the same number of genes ranked solely by absolute expression in craniofacial samples we observed gene ontology enrichments for general cellular and metabolic processes and distinct disease enrichments including Diamond Blackfan anemia and carcinomas (Figure S2A).”. The multi tissue comparison that we have constructed using the gini index is thus able to efficiently identify tissue-relevant genes across all tissues interrogated. Additionally we have not performed similar analyses in mouse and identified a subset of genes with conserved specificity in craniofacial development.

3. They validate that their analyses by finding enrichment of genes and regulators known from the literature to be relevant to human and mouse craniofacial development, but they do not perform any functional studies. Although their data analyses suggest novel human craniofacial candidate genes involved in embryonic development and disorders, they do not prove that any of these novel genes indeed are involved. This would require at a minimum either in vitro cell culture, ex vivo organoid, in vivo animal functional experimentation or detection of pathogenic variants associated with craniofacial conditions for which no disease gene has been previously assigned. There are WGS datasets of patients with craniofacial abnormalities including cleft palate available through the NICHD Gabriela Miller Kids First Pediatric Research Program that could be analyzed for variants in their candidate genes. Thus, their findings are generally descriptive and qualitative.

The reviewer is correct that we have not performed functional studies but we feel this is well beyond the scope of this work as we have implicated several hundred new candidates. We have taken the reviewer’s excellent suggestion to investigate genome sequencing data generated by the NICHD Gabriela Miller Kids

First Research program. In collaboration with the Leslie lab at Emory University we have demonstrated that our prioritization strategy is very significantly enriched for genes with de novo mutations in OFC trios. This new analysis provides strong support for 49 genes identified here and in those cohorts. This is reflected in the section “Regulatory hubs in co-expression networks reveal putative driver genes for the developing face”, paragraph 9, “To test this hypothesis, we inspected data from these 756 trios along with an additional dataset of 435 trios with cleft palate (CP). We identified 875 genes with at least 1 DNM and 58 genes with multiple DNMs in probands across this cohort. These could be further divided into genes with at least one DNM in probands with cleft lip with or without cleft palate (CL/P) and those with cleft palate only (CPO) (572 and 375 genes with 1 DNM, 29 and 21 genes with multiple DNMs respectively). We then interrogated these gene sets for significant overlaps with gene lists that we have assembled here, specifically CF Gini genes, CFSE targeted genes, hub genes, and prioritized genes. In contrast to findings in ASD, we did not find any significant overlaps between any of the DNM gene sets and hub genes for individual modules from WGCNA or hubs as a total. For CF gini genes we only found significant enrichment for genes with a DNM in CPO. For genes putatively targeted by at least one CFSE we found significant, but small enrichment for each category of all DNM genes, although the only category of multiple DNMs enriched was for CL/P. but only genes with multiple DNMs in CL/P. Surprisingly, we found significant overlaps with greatest fold enrichments with our prioritized genes and all the DNM gene sets. For each OFC grouping, the genes with multiple DNMs were more significantly enriched in our prioritized genes than those with a single DNM (Figure 6E, Table S7).”.

We have also engaged with the Scott lab at Baylor to analyze patients with mutations and copy number variations in EBF3. This is reflected in the same section, paragraph 5, “Haploinsufficiency of EBF3 was reported by multiple groups to cause hypotonia, ataxia, and delayed development syndrome, but orofacial clefting was not reported as a consistent feature (HADDs; MIM: 617330)109-114. A search of the exome sequencing clinical database at Baylor Genetics revealed 15 individuals whose phenotypes are likely to be explained by deleterious variants in EBF3. Three of these individuals had cleft palate. The first individual was a previously published 13-year-old female with a submucosal cleft palate and a short uvula111. She carried a de novo likely pathogenic c.512G>A, p.(G171D) [NM_001005463.3] variant in EBF3. The second was a six-year-old male with a cleft soft palate who carries a pathogenic de novo pathogenic c.487C>T, p.(R163W) [NM_001005463.3] variant in EBF3. He was born prematurely and had intrauterine growth restriction. Polyhydramnios was noted during pregnancy. His other phenotypes include delayed speech and language development, delayed motor milestones, hypotonia, flexion contracture, a positive Gower sign, strabismus, nasolacrimal duct obstruction, atrial septal defect, patent foramen ovale, obstructive sleep apnea, laryngomalacia and macrocephaly. The third was a five-year-old female with a cleft palate who carries a de novo likely pathogenic c.661G>A, p.(V221M) [NM_001005463.3] variant in EBF3. She was also noted to have delayed speech, intellectual disability, hypotonia, epicanthal folds, micrognathia, camptodactyly, hyperextensibility.”

With these inclusions, we believe that we have very strong evidence that this gene is a risk gene for orofacial clefting in humans. It has already been demonstrated that mouse mutants lacking EBF3 do not have such craniofacial problems. To address this we have now analyzed single nucleus gene expression data from more closely aligned timepoints across human and mouse, as indicated by our comparative Gini analyses from a timespan of mouse craniofacial development. This is reflected in the section “Single-nuclei RNA-sequencing of embryonic face”. In this fully reworked section we demonstrate that EBF3 and EBF2 are significantly coexpressed more frequently in human mesenchyme subtypes. Please see new figure 8.

Examples of other issues:

1. Title does not reflect the integration of chromatin analyses for enhancers, transcription factors, etc. that were performed.

We have changed the title to reflect the more comprehensive analysis that we have performed in this work.

2. Page 1, references cited, but not in reference list: NIDCR, 2018; OMIM, 2021.

We apologize for errors in referencing formats. In the conversion process between formats some of the references were not translated properly. We have hopefully corrected all such issues in the revised version.

3. Page 2, para 1, line 8: delete period after variation.

This has been rectified.

4. Figure S1A is not labeled as such.

We have added this label.

5. Figure S1B shows that one of the CS13 replicates is an outlier from the other CS13 samples, while all other samples cluster with their respective replicates. Why do the CS13 samples vary in gene expression profiles more than the other samples and is that of concern for further analyses?

We agree with the reviewer that one CS13 sample is a bit of an outlier when only comparing the primary samples. The PCA in Figure S1B is the initial projection without any batch or surrogate variable analysis. However when we expand these comparisons to CNCC and perform surrogate variable analysis these differences are much less pronounced (Figure 1B). In our multi-tissue comparison these differences are minimal in PCA space (Figure 1D). The gini analysis combines all the samples so this one sample is likely to have a small effect on the average gene expression in craniofacial samples (n = 12). In our pairwise differential expression analyses all of the samples cluster well by stage in the heatmap presented in Figure 4A. The outlier sample could be a slightly earlier sample from the ~2 to 3 day window that Carnegie stages encompass than the two other CS13 samples given the dendrogram in that figure. However we are pleased that primary tissues samples with so many potential variables demonstrate such consistent gene expression within timepoints and across development. Primary tissues from this timepoint are extremely rare and we do not have any others available for replacement. Without this sample we would be unable to perform differential expression analyses with CS13 timepoint and likely would not be able to construct the WGCNA network as it is recommended to have at least 15 samples.

6. Page 3, line 4 from the bottom, Figure S1D cited is not correct and should be Figure 1D.

This has been corrected

7. Page 4, para 2, what databases of CF abnormalities and cleft palate and analyses were performed to determine that ...239 genes demonstrated the highest average expression in our craniofacial samples ...were most enriched for CF abnormalities and cleft palate...also should be rewritten by adding words ...they were most enriched for genes implicated in CF abnormalities and cleft palate.

- We have provided lists of all genes obtained from Disgenet and CleftGeneDB in Table S2. We have also adjusted the language in this section as suggested.

8. Page 4, heading Genes expressed...abnormalities, para 3, lines 4 and 5, superscript font should be used instead of caret and Gou et al., 2015 should not be italics.

We have added appropriate use of subscript throughout and updated reference formats.

9. Page 4, heading Genes expressed...abnormalities, para 4, line 3, midline facial hypoplasia is not a correct phrase to describe facial features of Stickler syndrome. They should use midfacial hypoplasia. There is an extra parenthesis after Kondo et al., 2016.

We have changed this phrasing as suggested.

10. Page 4, heading Genes expressed...abnormalities, para 4, lines 5-8, they incorrectly state that Russell Silver was underappreciated in CF development. If they look at the Clinical Synopsis section of the OMIM entry there are several facial features (macrocephaly, delayed closure of anterior fontanel, large anterior fontanel, triangular face, prominent forehead, micrognathia, retrognathia, downturned corners of mouth, thin lips) noted for this condition that are well known to clinicians.

- The sentence specifically said “This gene has been linked in OMIM to Silver-Russell Syndrome patients and reported in a pygmy mouse phenotype, but is relatively underappreciated in CF development “.
- We do not imply that Silver-Russell Syndrome does not have distinct craniofacial features. Instead it is not clear that HMGA2 is a direct cause of SRS or any other craniofacial abnormalities. Only a handful of patients with Silver-Russell Syndrome have been reported to have copy number variants in the region encompassing HMGA2 in OMIM. Furthermore an “Expert Consensus Document” in Nature Reviews Endocrinology titled “Diagnosis and management of Silver–Russell syndrome: first international consensus statement” there is no specific mention of HMGA2 as a molecular cause. Instead the article focuses primarily on IGF2 and CDKN1C at the chr11p15 locus, chr14q32, and other loci not containing HMGA2. Thus it is clear that HMGA2 is underappreciated amongst experts as a cause for SRS or other craniofacial abnormalities.

11. Supplementary tables are not labeled as Table S1, S2, etc. so it is difficult to figure out the relevant datasets being cited throughout the manuscript. Is Table S1 available?

We apologize that the file names were obscured by the journal submission system. We have added these names to the readme sheet in each file to help with navigation. Table S1 was the sole table that is provided in simple format directly in the manuscript. We have updated that table to indicate the source of the CS22 samples from FACEBASE and the additional CS17 samples included in the revision. We have also detailed all mouse samples used in the new gini analysis in mouse.

12. Page 6, para 2, lines 3-4, expression profiles of their differentiated CNCC samples do not cluster with other published CNCC from the literature (Prescott et al. , 2015) why is that?

We view this in a different way. The CNCC data generated here cluster much better with the primary tissue than those generated in Prescott et al. We retrieved those data using recount2 to ensure that all processing and extracting of counts is the same as data we generated here. Thus it is unlikely to be a technical artifact. The protocol in the Prescott et al paper is quite distinct from the one employed here and we have shown at the chromatin level in Wilderman et al 2018 that these CNCCs are considerably different from the primary tissue as well. The differences are thus likely to reflect differences in the identity of the cells at the time of harvest. While it might be interesting to explore the differences of these two differentiation systems this is beyond the scope of the work presented here.

13. Page 6, para 3, lines 2-3, Figure 4A should be cited.

This figure has been cited here as suggested.

14. Page 7, full para 3, line 3 and elsewhere in the text, superscript font should be used instead of caret.

This has been corrected throughout.

15. Single-nuclei RNA-sequencing section was very descriptive about only major cell populations. They do not perform spatial-temporal analyses.

We have expanded our discussion of subtypes of mesenchyme. We only have a single timepoint included and these are disaggregated tissue samples so spatial-temporal analyses are not possible. We describe this in the manuscript in the section “Single-nuclei RNA-sequencing of embryonic face”. A more expansive study of comparative, cross-species single nucleus gene expression is beyond the scope of this investigation and will be part of a separate manuscript.

16. References list: articles do not have consistent format and other concerns such as Consortium, G. and GTEx Consortium (2017) has incorrect authors listed; Hansen et al. has year in incorrect order; Hsieh et al. does not have volume or page numbers; Lopes et al. and Reynolds et al. do not have page numbers; Taigman et al. may have incorrect authors' names.

We have corrected the reference formatting throughout.

Reviewer #3 (Remarks to the Author):

Yankee et al 2022, carry out an original investigation examining gene expression in human samples across key timepoints in craniofacial development. Bulk RNAseq is performed on tissue from different embryonic stages as well as from cranial neural crest cells derived from human embryonic cells, and this data is analysed for specificity of expression, overlap with disease and intersection with craniofacial specific regulatory elements. They demonstrate through differential gene expression analysis that there are dynamic expression changes across time, and clusters of genes are linked to GO terms, disease enrichment and chromatin states. The authors also used a gene co-expression analysis to identify modules of genes from the bulk RNAseq dataset. These were then characterised by GO analysis, expression trajectories and intersection with relevant gene lists to identify enrichments related to morphogenesis, disease and CF-specific expression. Candidate disease genes, including MAB21L1 and EBF3 were highlighted, supported by phenotypic observations in recently identified human disorders. The latter gene was further investigated in terms of gene regulation and linked to two GWAS clefting hits. Finally, single nuclei RNAseq analysis of craniofacial tissue is used to assess cell-specific expression of EBF3 and the correlating network gene EBF2.

This is a comprehensive analysis of difficult to study human embryonic material, that sheds significant light on gene expression during key stages of craniofacial development. The authors have considerable expertise in the bioinformatic analysis and integration of datasets to extract meaningful information and predictions of genes and loci that are candidates for craniofacial disorders. Altogether, this work adds an important resource for the field, and provides a basis for ongoing investigations into craniofacial disorders relevant to both developmental biologists and clinicians.

We thank the reviewer for recognizing the importance of the data generated and the extensive analysis we originally performed. We have built upon this based on the suggestions below to include more mouse data, more comparable snRNA-Seq data, and new analysis of orofacial clefting trio data. We hope this addresses all the weakness and suggestions indicated below.

I have a few considerations, and also minor comments below:

1) Tissue description. I think it is important that the tissue sampled is fully described. Does it correspond to the yellow highlighted regions in Fig1A? This isn't made clear. In general, this is important for the quality of the comparisons that are made between tissues and datasets. For example, was the tissue isolated for snRNAseq at CS20 equivalent to material isolated for bulk RNAseq? Similarly, does this apply to tissues used previously for epigenomic analyses?

We apologize for not better conveying the tissues used. We have added details to the text to better describe what was used and connect to figure 1 illustrations. Given the source of these samples we do not receive entire embryos. We only receive what has been identified by HDBR and flash frozen. We do our best to remove regions that are clearly not craniofacial but we do not know exactly what is included in every sample due to the lack of many other tissue landmarks. We have also updated Table S1 for all samples used in this study. This includes if tissue from the same embryo was previously used for epigenomic analysis in Wilderman et al 2018. We have removed CS20 data and replaced it with CS17. This directly overlaps with the bulk data

we have collected and we have now shown that this timepoint is most comparable to mouse E12.5 craniofacial tissue. This is elaborated on in the section “Genes expressed specifically during craniofacial development are enriched for craniofacial abnormalities”, paragraphs 5 and 6.

I think there are also issues comparing between human and mouse:

snRNAseq, CS20 versus E11.5. What is the evidence that these are the same stage for comparison? Also, the tissue isolated for the mouse E11.5 data is from a specific CF region, the lambdoidal junction. This is a small part of the overall craniofacial region. Is it viable to make comparison to this data when the tissues are likely not equivalent in time and space?

The reviewer raises excellent points about this cross species comparison. We have removed the CS20 data altogether and included CS17 samples instead. These samples directly correspond to samples used for bulk gene expression. We have also generated new data from mouse at E12.5. We demonstrated with bulk expression data that this timepoint is most similar to human CS17. We collected tissue from these mice to reflect the tissue available in human and did not perform any selection for lambdoidal junction as in the previously published E11.5 data. We have demonstrated that while the effect is not as striking as in the previous comparison, there are still distinct EBF2/EBF3 expression differences within the mesenchyme across species. This is elaborated on in the section “Genes expressed specifically during craniofacial development are enriched for craniofacial abnormalities”, paragraphs 5 and 6.

“To further determine if these CF expression biases reflected tissue and developmentally relevant differences, we asked whether such trends were also observed in mouse. To address this, we retrieved gene expression data uniformly processed by recount369 from developing mouse craniofacial prominences along with 20 other tissues ranging from embryonic day 10.5 (E10.5) to postnatal week 4, totalling 500 samples (Table S3). Based on PCA we found these data largely demonstrated expected differences in global gene expression that reflected their developmental origin (i.e. mesoderm, ectoderm, endoderm) , similar to our findings in the human datasets (Figure S2C). When we compared gene expression of orthologous genes across human and mouse craniofacial samples, we found that the second principal component oriented samples in a clear developmental trajectory (Figure S2D).

While it is difficult to directly compare developmental stages across species with dramatically different in utero development times, calculation of sample distances based on expression of all one-to-one orthologous genes indicated CS13 samples were most similar to E10.5 while CS17 samples were most similar to E12.5 (Figure S2E). When we performed the same Gini analysis as above for these mouse tissues, we found similar enrichments of coherent biological functions amongst tissue-biased genes (Figure S2F). For instance, heart biased genes were enriched for functions related to muscle cell differentiation and sarcomere organization while craniofacial biased genes were enriched for skeletal development and olfactory receptor activity.”

2) The snRNAseq data analysis feels under exploited and like a skim of the information that must be within this dataset, and its a bit strange that it is used simply to investigate EBF2/3 expression. I think that could be better accomplished through in situ analysis of comparable human and mouse craniofacial tissue.

We have expanded our exploration of the CS17 single nucleus data to identify global cell types and subtypes of mesenchyme. As described above, we have also co-projected this data with the appropriate cognate timepoint in mouse (E12.5). We have performed quantitative analyses between species to confirm our findings for EBF2/3 and also identified additional genes that may have species biases. Unfortunately the samples we receive are not readily applicable for sectioning. We have added discussion of these issues and our motivation for using single nucleus approaches. The section “Single-nuclei RNA-sequencing of embryonic face” has been fully reworked to reflect these re-analyses and edits, as reflected in Figures 7 and 8.

I don't find the network enrichment analysis in relation to the snRNAseq data convincing, without in situ validation of selected genes. The mesenchyme cluster (the vast majority of the cells) is likely composed of multiple individual subtypes so stating that there is enrichment for specific cell types would perhaps be better supported by subclustering of the mesenchymal cell population.

This was an excellent suggestion. We have performed additional subclustering of mesenchyme which reveals additional module specificity for subclusters. We detail these findings in the results in Figures 7 and 8, as part of the section "Single-nuclei RNA-sequencing of embryonic face" on pages 12-13.

TWIST1/PRRX2 are stated to have similar expression in mesenchyme between species, however, from the data presented this is really hard to judge. If the mesenchyme was subclustered, and expression correlated with the top marker genes for each subcluster, this might help to assess these correlations. The same is true for the module score analysis, where there do appear to be zones of expression, particularly in the mouse mesenchyme.

We have removed this comparison in order to better explore the overall CS17 data.

In the human snRNAseq data EBF2/3 appear to be expressed in a subset of mesenchymal cells. What are these? There is some expression in the mouse cells – are these the same cells? Again, subclustering might clarify this.

This was again an excellent suggestion. We have used better aligned cell types and more similar data generation to make a more accurate comparison. We have also performed subclustering of mesenchyme to identify subclusters with strongest differential expression across species. All these changes are reflected in the newly reworked section "Single-nuclei RNA-sequencing of embryonic face", in figures 7-8.

It doesn't look like there is much expression of Ebf2/3 in the E11.5 data, but I'm not sure its informative to compare to the E13.5 Allen data to confirm expression patterns, which is 2 days later in development (even then I think there is mesenchymal expression at E13.5). It would be better to have in situ analysis of Ebf2/3 at E11.5 (or equivalent stage to CS20). Overall, the data only suggest differences in expression, which might actually be due to differences in the tissue sampled, and the stage of development.

We have tried to simplify the comparisons based on the reviewers suggestions, as detailed above, to CS17 and E12.5. While no cross species comparison will ever be perfect, we feel this is the most reliable cross species comparison that is driven by quantitative comparisons of bulk tissue. We have added human genetic support for EBF3 in this new version to better support our argument that this gene is a risk factor for orofacial clefting in humans. This is reflected in the section "Single-nuclei RNA-sequencing of embryonic face", and figures 7-8.

This was a multiome analysis. It would be interesting to look at correlations between open chromatin and specific gene expression. Do the mesenchymal (subsets) ATACseq peaks correlate with the other epigenetic data in candidate genes and their expression? How does the data relate to the bulk CS22 data etc

Unfortunately the ATAC side of these specific multiome experiments failed. The reagents used for these experiments were the first versions provided by 10X Genomics and seemed to have had some technical issues. The single nucleus RNA experiments passed all QC checks recommended by 10X and as we demonstrate with comparisons to mouse samples collected here and from published literature are very comparable. We are very interested in this idea and will be exploring in a future study for the newest multiome data that has been successful.

3) Care needs to be taken in discussing mouse-human differences in EBF3 mutation outcome and concluding that that's there are different functions or expression patterns. The mouse mutants do not have overt craniofacial defects, but the phenotype in humans is not that strong in terms of craniofacial findings. Subtle craniofacial effects can easily be missed in a mouse mutant. Conversely, there are clear overlaps in the mouse and human phenotype (eg Blackburn 2017).

We agree with the reviewer that the language of our discussion could be more precise. We have toned down these arguments and instead focused on what we can show in humans. We now provide new analysis of human genetic data to provide stronger support that *EBF3* is indeed a risk gene for orofacial clefting beyond the subtle phenotypes described in HADDS. These findings are detailed on pages 10 and 11.

4) Mesoderm also contributes to craniofacial development, was this seen in the data? CNCCs are rightly used in the study, with a significant overlap with CS13, but I wondered if there should be a comment relating to mesodermal involvement.

While this is an interesting point we do not have bulk gene expression data available specifically from mesoderm from human embryonic samples to address this. By CS17 there does not seem to be a cell population that maintains that identity in our analyses. We will be generating single cell data from CS13 samples that are likely to harbor such cells but this will be addressed in a separate study. We have added a statement to the discussion to acknowledge that we cannot exclude involvement of other lineages to craniofacial development (see page 15 lines 20-24).

Minor points

Line 93/94 – As pcw is used in the summary, but Carnegie stages used subsequently, it would make things clearer if the equivalent pcw range is included here at the start of the results.

We apologize for the confusion and have added these descriptions early in the results on page 3, section “Characterization of global gene expression during human craniofacial development”, paragraph 1.

Line 95 – these same stages – not CS22?

We have detailed in Table S1 exactly what samples have been used in this study and in Wilderman et al. We also detail the source of the CS22 data being from FACEBASE.

Line 98 – should reference to Figure 1B be here?

This has been updated.

Line 100 – why is one of the CS13 samples so different in the staging PCA plot?

We have addressed this issue in response to other reviewer above.

Line 133 – Figure 1D rather than S1D?

This has been updated.

Line 151 – what are the 239 genes, and what genes are within each disease category? List in table S2?
We have added these lists and Disease ontology enrichment to table S2

Line 158 –the transcription factor enrichment – can these genes also be listed in table S2? Similarly, what are the genes within each category?

We provide table S2 with a prefiltering of genes for craniofacial bias. We now also include genes from databases and TF identities that were used in our analyses as separate tabs in table S2.

Line 164 – S2A should be S2. The Figure S2 figure legend – disease(?) categories

We have updated Figure S2 and addressed these citations appropriately. We have also updated the legend for this figure to address this issue.

Line 196 – source of embryonic heart data? What stages were used?

These are data we published previously in Vanoudenhove et al 2020. We have more clearly referenced in the text to indicate the source of the data.

Line 202 – I was confused as to why Fig 1D is referred to here. Is the enrichment data shown anywhere?

We apologize this should have been a reference to Figure 3A.

Line 206 – insert reference to Figure 3B. Fig S3A,B?

We apologize for this omission, we have added these references here as suggested.

Line 234 – CFSE track for MSX2 is not shown

We have updated this figure to include these regions.

Line 266 – progressive dampening over developmental time – expression does seem to come back on in CS22

We agree with the reviewer that some genes do seem to be reactivated at CS22 for most genes in this specific cluster they remain lowly expressed compared to the early time period we defined.

Line 273 – move S6E reference to line 276

We have updated as suggested.

Line 281 – The PAX3 and SOX9 trajectories don't look that different from the overall trajectory

We agree with the reviewer and were not precise in our description. We have specifically only mentioned PAX7 and RXRG that are clearly different from the overall Dark Green cluster trajectory.

Line 299 - , suggesting

Line 323 – 29 modules, but only 17 shown in Fig 5A?

For simplicity we only showed the modules with significant gene ontology enrichments. We have updated the legend and text to reflect this difference.

Line 397 – in Fig 6C the diamond shape in the inset box is unclear

The conversion process seemed to make that shape unclear. We have updated this figure to address this issue.

Line 405 – references for these previous associations?

These references were cited in an awkward location and have been moved to this location as suggested.

Line 413 – paralog?

Sorry for this error, paralog is more specific in this context.

Line 415 – Why was CS20 chosen, and not one of the timepoints used in the RNAseq analysis?

This is a major point that we have addressed in this revision. We have removed this data and replaced with CS17. Please see our detailed responses to the other reviewer on this issue.

Line 415 - How many nuclei were isolated/sequenced from the 2 samples?

Line 416 - What were the marker genes used for clustering into the 6 main cell types?

These points have been addressed for the newly added CS17 and E12.5 data.

Line 427 – FigS9C -> FigS11C?

Sorry for this error. FigS11C is the correct reference.

Line 429 – Please give more information on the module score – how its calculated (methods) and what it shows. Enrichment does seem to be in the mesenchymal population, but how does that look when these populations are subclustered to reveal the different constituent populations?

The module score has been described in the methods and is an in built function of Seurat. Now that new samples have been added we have provided the analysis of subcluster mesenchyme.

Line 433 – MAB21L1/Mab21I1 expression cannot be described as cell-type restricted – it appears to be expressed in multiple cell types in both human and mouse.

We have removed this section altogether and focused primarily on EBF3/EBF2 in the new CS17/E12.5 comparisons.

Line 434 - FigS9D -> FigS11D?

This has been corrected

Line 435 – Are these really comparable populations of cells (both spatially and temporally) to state that there are striking differences?

We do not have spatial information in these data. We are not directly co-projecting human and mouse data allowing us to make quantitative comparisons at each cluster and subcluster across species.

Line 439 – What human stage of Cf development does E13.5 correspond to? The scRNAseq data is from E11.5, perhaps expression patterns have changed?

Our responses above address the issue of heterochrony across species. We have established in our new analysis that CS17 and E12.5 are most comparable.

Line 441 – TWIST1 and PRRX2 in Fig 7D look to me to be expressed in different cell types within the mesenchymal cluster, with some overlap in the PRRX2 domain.

We have removed this and instead focus on EBF2/3 comparisons across species.

Line 444 – lowercase for mouse genes in Fig S11E

We have removed this figure altogether in the revision.

Line 444/445 – There does appear to be Ebf3 in craniofacial mesenchyme. What about Ebf2? The statement that the data suggests the 2 genes have been removed or gained in humans is too strong

We agree that this was too strong of a statement. Our new analysis confirms cross-species differences in mesenchyme subtypes and our mouse WGCNA analysis revealed they are in different modules than the majority of genes from the human black module. While this does suggest different roles we cannot say they have been removed from the program. We have adjusted the language here and the discuss to reflect these findings.

Line 480 – Fig4C?

This has been corrected to figure 4A

\

Line 481 – changes over time?

This has been added

Line 496 – genes with mutational constraint?

This was indeed awkward phrasing and we have changed as suggested.

Line 511 – MGMT previously stated to be broadly expressed in the data? This also appears to be the case in Fig3D...

We have added this reference and we also mention the low gini score that demonstrates expression across most human tissues.

Line 770 – Include a description of the dissection describing exactly what tissue was analysed

We have added more language here and in the results to describe what tissues are obtained.

Line 798 – Is this exactly the same tissue as analysed in the bulk RNAseq? There is minimal information on the protocol used. Were the 2 samples treated at the same time? Were there technical replicates? When were the samples combined? Presumably they were separate – do they overlap in UMAP plots?

We have replaced these data with CS17 and E12.5. We have coprojected these in new Figure 7 and 8 and show that replicates and species cells have similar numbers and proportions.

Line 856 – Figure S1F?

Line 883 – Table S2?

Line 900 – For Figure 3B, we used..

Line 916 – Included the....?

We have fixed this sentence to indicate that all samples from these stages were included in the surrogate variable analysis.

Table S1 – Where is CS22 for bulk RNAseq?

We have added a notation that these data were obtained from FACEBASE.

REVIEWER COMMENTS

Reviewer #1 (Remarks to the Author):

This is a revision of an interesting and comprehensive manuscript that generated and analyzed bulk and single cell RNA-seq on human craniofacial tissues from embryonic development 4 to 8 post conception. They perform multiscale analyses integrating their RNA-seq data with human and mouse GWAS, WGS, dbGaP, ChIP seq and Roadmap Epigenome datasets.

The major additions to this revision addresses conservation of expression of craniofacial genes between human and mouse and identification of variants in human craniofacial candidate genes associated with oral clefting. Specifically, the authors performed the following new studies:

1. Extensive analysis of expression of mouse craniofacial genes during embryonic development and identification of conservation of craniofacial genes.
2. Comparison of human CS17 and mouse E12.5, craniofacial gene expression. However, mouse E13, not E12.5 is considered the analogous developmental stage to human CS17.
3. Validation of their network analyses by detection of pathogenic variants in genes associated with craniofacial conditions. They now use ES/GS datasets from patients with craniofacial abnormalities including cleft palate available through the NICHD Gabriela Miller Kids First Pediatric Research Program and DECIPHER to analyze for variants in their candidate genes.
4. Prioritization strategy of genes enriched for de novo mutations associated with oral clefting.
5. Identification of the human candidate gene, EBF3 from their expression studies, and confirm it is an oral clefting risk gene.

There are some wet lab approaches that they do not apply to validate your bioinformatics analyses:

1. The samples and tissues obtained are not necessarily dissected specifically to implicate genes expressed in specific structures related to specific disease types such as oral clefting versus craniosynostosis. They do not perform spatial expression studies by in situ or immunohistochemistry to validate that genes are specific to cell types or tissues to show that these genes indeed express in the specific structures such as the developing palatal shelves, etc. as predicted by their bioinformatics analysis.
2. They do not perform any functional studies using in vitro cell culture, ex vivo organoid, or in vivo animal experimentation.

Examples of minor issues:

1. There is a "7" affiliation, but no author is assigned that affiliation.
2. Summary states they use exome sequencing data, but they also use genome sequencing data.
3. There is no mention of the expression of the TWIST1 gene, shown in Figure S1E, in the text. Also this gene expression seems to be very different in craniofacial tissues compared to CNCC, so these data seem contrary to the authors' statements that gene expression is similar between craniofacial tissues and CNCC.
4. In the text line 145, Figure S1F is mentioned, but there is no panel F provided in the supplemental figures.
5. In the text line 174-179, it is unclear why it is useful that "...the Gini index analysis, using the same number of genes ranked solely by absolute expression in craniofacial samples we observed gene ontology enrichments for general cellular and metabolic processes and distinct disease enrichments including Diamond Blackfan anemia and carcinoma..." Why are these particular enrichments important or relevant to craniofacial tissues or diseases?
6. In text line 224, Figure 2D is mentioned, but there is no panel D or legend provided.

7. In text line 242-243, Roadmap Epigenome reference is not in proper formatted with a numerical citation. Several other cited references do not use numbers, but have the authors names e.g., Bishop et al. 2020.

8. References list: as noted in the original submission articles do not have consistent format. Some titles of articles are all upper case and some not. Other examples, reference 14 has no authors listed. Reference 7 should have a space between “for” and “Studying”

Reviewer #3 (Remarks to the Author):

The authors have very much improved their manuscript, addressing the vast majority of the reviewer comments. This includes the introduction of new data from analysis at CS17 and alignment with new mouse E12.5 data. This is a significant addition and revision. The single nuclei analysis is much more thorough, informative, and convincing. In addition, the authors have added human genetics data to the paper, substantially strengthening the association of candidate genes, particularly EBF3, with human craniofacial pathogenesis.

On a minor note I found the last paragraph of the results relating to fig8c a little confusing: the human-biased expression for mesenchyme cluster 3 - there doesn't seem to be any expression in this cluster. Similarly "Ebf2 was mouse biased in cluster 6".

We would like to thank the reviewers for their time considering our manuscript and please that both recognized the substantial work we included in the previous revision. They raise no major issues with our new analyses and only have minor issues that we address below. We also provide a version of the manuscript that highlights all changes in this revision compared to the previous submission.

REVIEWER COMMENTS

Reviewer #1 (Remarks to the Author):

This is a revision of an interesting and comprehensive manuscript that generated and analyzed bulk and single cell RNA-seq on human craniofacial tissues from embryonic development 4 to 8 post conception. They perform multiscale analyses integrating their RNA-seq data with human and mouse GWAS, WGS, dbGaP, ChIP seq and Roadmap Epigenome datasets.

The major additions to this revision addresses conservation of expression of craniofacial genes between human and mouse and identification of variants in human craniofacial candidate genes associated with oral clefting. Specifically, the authors performed the following new studies:

1. Extensive analysis of expression of mouse craniofacial genes during embryonic development and identification of conservation of craniofacial genes.

We are glad the reviewer recognizes the substantial effort to perform these analyses and that the systematic identification of conserved craniofacial genes is well received.

2. Comparison of human CS17 and mouse E12.5, craniofacial gene expression. However, mouse E13, not E12.5 is considered the analogous developmental stage to human CS17.

There were no mouse E13 craniofacial RNA-Seq data sets available in our exhaustive search of Recount3. We demonstrated in our global bulk-gene expression analysis that CS17 and E12.5 were the most similar stages for which we had public data. Furthermore, as far as we are aware, this is the first work to systematically compare gene expression directly between human and mouse craniofacial tissues across multiple stages. Therefore we focused on timepoints that we had established as most similar at the global expression level for single nucleus RNA-Seq experiments to address this reviewer's original objection. While it is possible that CS17 and E13 are most similar, the reviewer does not provide a reference and we did not have data to address such a question.

3. Validation of their network analyses by detection of pathogenic variants in genes associated with craniofacial conditions. They now use ES/GS datasets from patients with craniofacial abnormalities including cleft palate available through the NICHD Gabriela Miller Kids First Pediatric Research Program and DECIPHER to analyze for variants in their candidate genes.

We thank the reviewer for requesting this analysis and we feel it has greatly strengthened the conclusions from our work.

4. Prioritization strategy of genes enriched for de novo mutations associated with oral clefting.

The suggestion to include Gabriela Miller Kids data in our analysis was a particularly excellent one. This allowed us to show the power of our systematic approaches and provide the field with new genes to focus their efforts.

5. Identification of the human candidate gene, EBF3 from their expression studies, and confirm it is an oral clefting risk gene.

We thank the reviewer for the suggestions to further validate our work with human genetics. This has resulted in a new collaboration and hopefully demonstrates there are likely many other human craniofacial disease risk genes that have distinct roles in mice.

There are some wet lab approaches that they do not apply to validate your bioinformatics analyses:

1. The samples and tissues obtained are not necessarily dissected specifically to implicate genes expressed in specific structures related to specific disease types such as oral clefting versus craniosynostosis. They do not perform spatial expression studies by in situ or immunohistochemistry to validate that genes are specific to cell types or tissues to show that these genes indeed express in the specific structures such as the developing palatal shelves, etc. as predicted by their bioinformatics analysis.

As we detailed in the resubmission, the samples that we have obtained are not complete embryos and have already had craniofacial tissue dissected. Thus they are not suited for current commercially available spatial gene expression studies. Furthermore, such samples are becoming increasingly rare and we feel that low information density in situ (of which only a small number have ever been performed on human embryos at these stages see <https://hdbratlas.org/gene-expression.html>) would not be an efficient use of the tissue.

2. They do not perform any functional studies using in vitro cell culture, ex vivo organoid, or in vivo animal experimentation.

We have performed comprehensive gene expression analysis of multiple stages of mouse and human craniofacial development and have validated our findings with many different public datasets in a systematic fashion. One of the clear examples we have identified as a disease risk gene is EBF3 and this has been demonstrated to have different outcomes in mice. Since a clear human craniofacial organoid system has not been established and vetted we feel such experiments are beyond the scope of the current work.

Examples of minor issues:

1. There is a “7” affiliation, but no author is assigned that affiliation.

This has been added as an affiliation for the corresponding author.

2. Summary states they use exome sequencing data, but they also use genome sequencing data.

We have removed “exome” so that it reads only “sequencing data” to be more inclusive.

3. There is no mention of the expression of the TWIST1 gene, shown in Figure S1E, in the text. Also this gene expression seems to be very different in craniofacial tissues compared to CNCC, so these data seem contrary to the authors' statements that gene expression is similar between craniofacial tissues and CNCC.

We apologize that an updated version of Figure S1 was not included in the supplement. The text in the resubmission refers to PAX7 which is in the updated figure and shows very similar expression in the CNCC system. We agree with the reviewer that TWIST1 is not robustly expressed in the culture model, however this is one gene and we go on to show across thousands of genes the CNCC system is similar to the primary tissue.

4. In the text line 145, Figure S1F is mentioned, but there is no panel F provided in the supplemental figures. As mentioned above we apologize for not including an updated figure S1 that contains a panel F

5. In the text line 174-179, it is unclear why it is useful that "...the Gini index analysis, using the same number of genes ranked solely by absolute expression in craniofacial samples we observed gene ontology enrichments for general cellular and metabolic processes and distinct disease enrichments including Diamond Blackfan anemia and carcinoma..." Why are these particular enrichments important or relevant to craniofacial tissues or diseases?

The reviewer's original criticism of the gini index was to whether it added much beyond just choosing genes highly expressed in craniofacial tissue. Specifically they asked "**Why not 250 craniofacial genes with the highest average of expression?**" When we did exactly this we get the results described showing there is NOT clear craniofacial relevance. The point is that the gini index reveals tissue specific biology and expression in a tissue without relationship to other tissues yields only general functions. We have added an additional statement to make this point: "Whereas using only gene expression strength in a single tissue identified genes with general, house-keeping functions."

6. In text line 224, Figure 2D is mentioned, but there is no panel D or legend provided.

We apologize that this figure was cut off in the resubmission. Panel D is now included in this figure.

7. In text line 242-243, Roadmap Epigenome reference is not in proper formatted with a numerical citation.

We have updated this citation.

Several other cited references do not use numbers, but have the authors names e.g., Bishop et al. 2020.

We have cited the Bishop et al. 2020 paper with numeric format when first mentioned but have used the expanded format when writing specifically about that work to prevent very awkward sentence structure.

8. References list: as noted in the original submission articles do not have consistent format. Some titles of articles are all upper case and some not. Other examples, reference 14 has no authors listed. Reference 7 should have a space between "for" and "Studying"

We have updated these two references. We have been using Refworks for all referencing and apologize that it is not consistent in its formatting. However all the references can easily be located in the literature and does not impact findings of our work.

Reviewer #3 (Remarks to the Author):

The authors have very much improved their manuscript, addressing the vast majority of the reviewer comments. This includes the introduction of new data from analysis at CS17 and alignment with new mouse E12.5 data. This is a significant addition and revision. The single nuclei analysis is much more thorough, informative, and convincing. In addition, the authors have added human genetics data to the paper, substantially strengthening the association of candidate genes, particularly *EBF3*, with human craniofacial pathogenesis.

We appreciate the reviewer acknowledging our significant efforts for revision and glad this important work will be available to the field.

On a minor note I found the last paragraph of the results relating to fig8c a little confusing: the human-biased expression for mesenchyme cluster 3 - there doesn't seem to be any expression in this cluster. Similarly "Ebf2 was mouse biased in cluster 6".

We apologize for the lack of clarity in describing the result. To be more clear we have excluded any subclusters of the mesenchyme where median gene expression was less than 1 in both human and mouse for significance testing and simplified the language. The section now reads:

"When we inspected co-expression of these two genes we found a striking bias toward human cells to jointly express them. This was particularly apparent in the mesenchyme (Figure 8B). These results in the mesenchyme were of particular interest due to this cell type's relevance to craniofacial phenotypes. Therefore, we examined the mesenchymal subclustering for expression of *EBF2/EBF3* across species in these subpopulations. For clusters with median expression greater than 1, we observed significantly human-biased expression for both genes in mesenchyme subclusters 0 and 1 (Figure 8C). In contrast we did not find significant mouse-biased coexpression of these two genes in any subcluster. *Ebf3* was biased toward mouse in subclusters 4 and 5, while *Ebf2* was not biased toward mouse in any subcluster."